# VLM-SubtleBench: How Far Are VLMs from Human-Level Subtle Comparative Reasoning?

**Minkyu Kim**[1,*]**, Sangheon Lee**[1,2,*,†]**, Dongmin Park**[1]
[1]KRAFTON, [2]KAIST

## Abstract

The ability to distinguish subtle differences between visually similar images is essential for diverse domains such as industrial anomaly detection, medical imaging, and aerial surveillance. While comparative reasoning benchmarks for vision-language models (VLMs) have recently emerged, they primarily focus on images with large, salient differences and fail to capture the nuanced reasoning required for real-world applications. In this work, we introduce **VLM-SubtleBench** [1] , a benchmark designed to evaluate VLMs on *subtle comparative reasoning*. Our benchmark covers ten difference types—Attribute, State, Emotion, Temporal, Spatial, Existence, Quantity, Quality, Viewpoint, and Action—and curate paired question–image sets reflecting these fine-grained variations. Unlike prior benchmarks restricted to natural image datasets, our benchmark spans diverse domains, including industrial, aerial, and medical imagery. Through extensive evaluation of both proprietary and open-source VLMs, we reveal systematic gaps between model and human performance across difference types and domains, and provide controlled analyses highlighting where VLMs' reasoning sharply deteriorates. Together, our benchmark and findings establish a foundation for advancing VLMs toward human-level comparative reasoning.

## 1 Introduction

The ability to discern *subtle visual differences*, i.e., minor discrepancies between otherwise similar objects, scenes, or situations, is central to human cognition. It enables us to perform a wide range of fine-grained comparative tasks, such as *recognizing micro-expression changes* in daily life (Li et al., 2022), *detecting anomalies* in manufacturing (Bergmann et al., 2019), *assessing subtle variations* in satellite imagery (Asokan & Anitha, 2019), and *distinguishing disease stages* in medical imaging (Litjens et al., 2017). Beyond everyday life, the capacity to detect and reason about such subtle differences has expanded human abilities toward higher forms of intelligence, underpinning advances from scientific discovery to complex social organization.

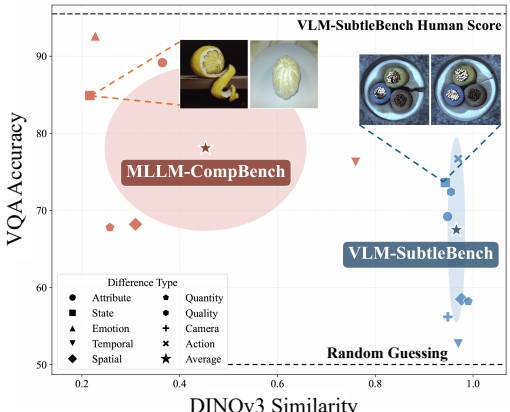

Figure 1: Comparison of VLM-SubtleBench and MLLM-CompBench with GPT-4o.

Recently, vision-language models (VLMs) have shown remarkable progress toward artificial general intelligence (AGI), demonstrating promising results in various tasks, such as visual question answering (VQA) and scene description (Zhang et al., 2024). Yet, most progress has primarily centered on single visual inputs, e.g., an image or a video, while comparative tasks that require comparison over

---

[*]Equal contribution.

[†]Work done during an internship at KRAFTON.

[1]Our dataset and code are available at `https://huggingface.co/datasets/KRAFTON/VLM-SubtleBench` and `https://github.com/krafton-ai/VLM-SubtleBench`.

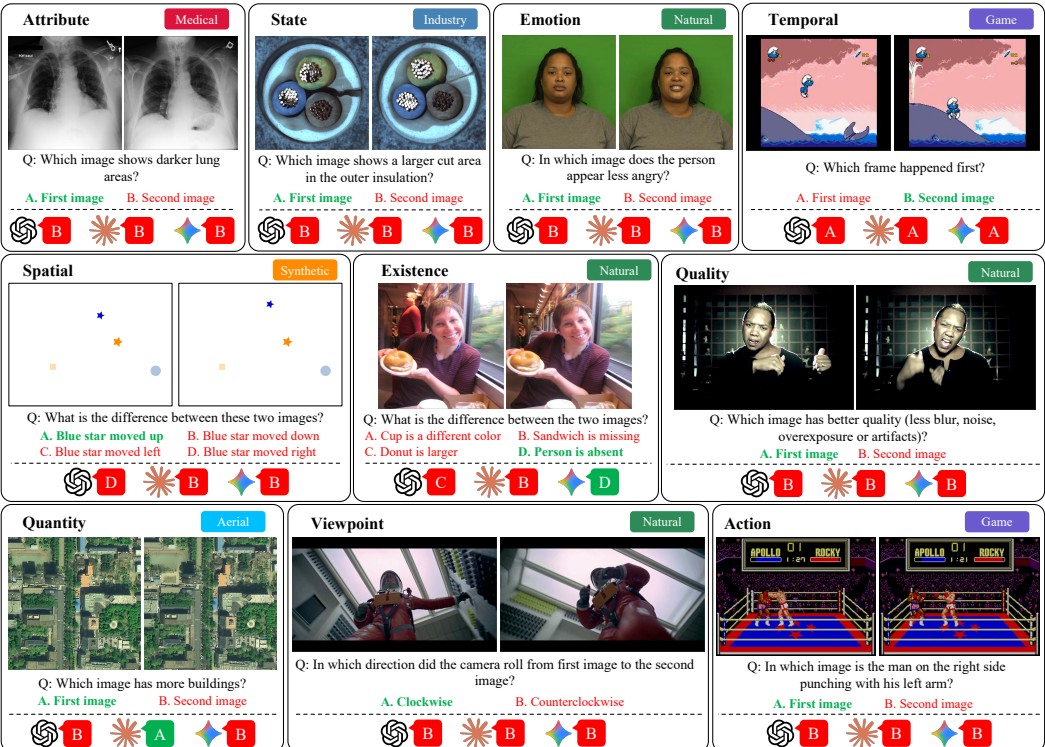

Figure 2: Example tasks from the VLM-SubtleBench, covering ten difference categories (Attribute, State, Emotion, Temporal, Spatial, Existence, Quality, Quantity, Viewpoint, Action) and six domains (natural, game, medical, industry, aerial, synthetic). For each example, the correct answer is highlighted in **bold green**. Model responses from GPT-5-main, Claude-sonnet-4, and Gemini-2.5-pro are shown beneath each question in order. Some VQA instances are simplified due to space constraints; full versions and additional examples are provided in the appendix.

multiple inputs, e.g., two images, have received relatively little attention. However, narrowing this gap is increasingly important, as recent applications often deploy VLMs as agents to perform complex tasks involving comparative reasoning across multiple observations, e.g., self-reflection over previously observed scenes (Hong et al., 2024). To truly serve as human-level surrogates, advancing VLMs' capacity for advanced comparative reasoning is becoming indispensable.

A few benchmarks have been presented to evaluate the comparative reasoning capabilities of VLMs. However, as shown in Figure 1, prior benchmarks focus on relatively simple comparisons between fairly dissimilar scenes, e.g., identifying differences in salient features (states) of distinct objects (two lemons), as indicated by the low average subtlety scores of MLLM-CompBench measured by embedding similarity with DINOv3 (Siméoni et al., 2025). As a result, they are easily solved by state-of-the-art VLMs, such as GPT-4o. In addition, most prior benchmarks are composed of natural images, and thus fail to assess performance in specialized domains such as industry or medical (See Table 1). Therefore, this calls for a new benchmark that can evaluate human-level subtle comparative reasoning across diverse domains and high-difficulty tasks.

To this end, we present VLM-SubtleBench, a benchmark designed to evaluate human-level comparative reasoning capabilities of VLMs. As illustrated in Figure 2, VLM-SubtleBench comprises 13K triplets of image pairs, questions, and answers, covering 10 representative difference types, i.e., *Attribute*, *State*, *Emotion*, *Temporal*, *Spatial*, *Existence*, *Quality*, *Quantity*, *Viewpoint*, and *Action*, collected from diverse image domains, i.e., *Natural*, *Game*, *Industry*, *Aerial*, *Synthetic*, and *Medical*. Even advanced proprietary VLMs such as GPT-5 and Gemini-2.5-pro exhibit significant performance gaps compared to humans, especially on difference types that demand spatial, temporal, and viewpoint reasoning.

We conduct systematic studies on the performance of open-source and proprietary VLMs, the effectiveness of test-time prompting strategies on comparative tasks, and controlled experiments with synthetic image pairs, which reveal the extent to which VLMs align with human performance and

Table 1: **Summary of Comparative Reasoning Benchmarks.** 'Is Subtle?' indicates whether the benchmark contains image pairs whose DINOv3 similarity averages at least 0.8, meaning the differences are subtle. VLM-SubtleBench is the *only* benchmark that focuses on subtle comparison, spans diverse domains, and includes both multiple-choice questions and captioning tasks.

| Benchmarks | Is Subtle? | # Domains | # Diff. Types | MCQ | Captioning |
|---|---|---|---|---|---|
| Birds-to-Words (Forbes et al., 2019) | ✗ | 1 | 1 | ✗ | ✓ |
| Spot-the-Diff (Jhamtani et al., 2018) | ✓ | 1 | 1 | ✗ | ✓ |
| MLLM-CompBench (Kil et al., 2024) | ✗ | 1 | 8 | ✓ | ✗ |
| **VLM-SubtleBench (Ours)** | ✓ | 6 | 10 | ✓ | ✓ |

where their reasoning sharply deteriorates. Our findings show that (1) proprietary VLMs still leave large gaps from human performance, particularly on spatial, temporal, and viewpoint reasoning where even the best model trails humans by over 30 percentage points; (2) simple prompting strategies, such as chain-of-thought prompting, grid layouts, and overlapping images, yield only limited improvements; and (3) VLMs are highly sensitive to difficulty factors such as object size and count. Together, VLM-SubtleBench and our experimental studies provide critical insights toward narrowing the gap between VLMs and humans in subtle comparative reasoning.

## 2 RELATED WORK

**Multi-Image Benchmarks for VLMs.** While most VLM benchmarks focus on single-image understanding (Liu et al., 2024b; Ying et al., 2024; Wu et al., 2024), a growing body of work evaluates multi-image reasoning. Early efforts such as NLVR2 (Suhr et al., 2019) and Winoground (Thrush et al., 2022) tested whether models can reason over image pairs through sentence verification and compositional matching, respectively. More recently, BLINK (Fu et al., 2024) evaluates core visual perception tasks such as relative depth and visual correspondence, exposing large human–model gaps on low-level visual skills. REMI (Kazemi et al., 2024) focuses on high-level relational reasoning, such as analogy and sequential pattern completion, across multiple images. MuirBench (Wang et al., 2025) provides a comprehensive suite of 12 multi-image task types, ranging from visual retrieval to scene understanding. While these benchmarks collectively advance multi-image evaluation, none are specifically designed to evaluate *comparative reasoning* between image pairs, identifying what has changed between two similar images. MLLM-CompBench (Kil et al., 2024) is the closest to our work, as it directly evaluates comparative judgments across eight difference types. Yet, its image pairs typically depict different subjects or settings with salient differences, as evidenced by their low average similarity scores (Figure 1). In contrast, VLM-SubtleBench focuses on subtle differences within nearly identical contexts, requiring genuine cross-image comparison (Table 1).

**Difference Understanding across Domains.** Detecting and describing visual differences has been studied across specialized domains. Spot-the-Diff (Jhamtani & Berg-Kirkpatrick, 2018) introduced textual descriptions of changes between surveillance frames, CLEVR-Change (Park et al., 2019) provided a synthetic testbed for change captioning, and Birds-to-Words (Forbes et al., 2019) collected fine-grained comparative descriptions of bird photographs. In the medical domain, MIMIC-Diff-VQA (Johnson et al., 2019) supports difference-focused VQA on chest X-ray pairs, while GeoBench (Lacoste et al., 2023) covers remote sensing tasks including change detection for Earth monitoring. While these works demonstrate the importance of difference understanding, each is confined to a single domain and a narrow set of difference types, leaving no unified benchmark that evaluates VLMs' comparative reasoning across diverse visual distributions.

**Image Difference Captioning.** A related line of work focuses on generating textual descriptions of image differences. Img-Diff (Jiao et al., 2025) constructs contrastive image pairs targeting object-level changes such as replacement and removal, OneDiff (Hu et al., 2025) aggregates multiple sources and trains a generalist model covering color, texture, and spatial changes, and DiffTell (Di et al., 2025) curates a high-quality dataset that further encompasses style and text manipulation. While these efforts advance difference captioning with larger-scale data and dedicated models, they primarily focus on natural images and the captioning task alone. Our benchmark complements this line of work by providing both VQA and captioning tasks across ten difference types and six domains, enabling a more comprehensive evaluation of comparative reasoning.

## 3 VLM-SUBTLEBENCH

In this section, we introduce **VLM-SubtleBench**, a benchmark designed to evaluate subtle comparative reasoning capabilities of VLMs. VLM-SubtleBench focuses on whether models can reliably identify **subtle differences** between two highly similar images, a key aspect of comparative visual reasoning. In the following, Section 3.1 describes the scope of the benchmark, covering the visual domains and difference categories included. Section 3.2 details the dataset curation process, and Section 3.3 explains how caption annotations were performed to ensure high-quality textual descriptions. Section 3.4 presents dataset statistics that highlight the diversity of VLM-SubtleBench.

### 3.1 SCOPE OF VLM-SUBTLEBENCH

**Covered Image Domains.** To evaluate whether a model possesses human-level subtle comparative reasoning across diverse, cognitively demanding tasks, it is essential to cover images from a wide range of domains. Thus, we design VLM-SubtleBench to span *six* representative image domains: **Natural Scenes**, capturing everyday real-world photographs (Abu-El-Haija et al., 2016; Lin et al., 2014; Souček et al., 2022; Cao et al., 2014; Livingstone & Russo, 2018; Kossaifi et al., 2017; Gupta et al., 2016; Zhou et al., 2025; Wang et al., 2024; Idrees et al., 2018; Lin et al., 2025); **Game Environments**, simulated yet realistic scenes that test generalization beyond natural images (Abu-El-Haija et al., 2016; Lin et al., 2025); **Aerial Imagery**, covering remote sensing and overhead views where subtle spatial differences are critical (Liu et al., 2024a; Huang et al., 2022); **Industrial Inspection**, representing structured settings where fine-grained defects or anomalies need to be detected (Bergmann et al., 2019; 2022); **Medical Imaging**, where diagnostic reasoning often requires distinguishing subtle changes across visits (Johnson et al., 2019; Hu et al., 2023); and **Synthetic Primitives**, consisting of abstract 2D shapes with varying colors and arrangements on plain backgrounds, which further allows controlled analysis.

**Covered Difference Types.** We also design VLM-SubtleBench to cover diverse types of differences. Specifically, we follow the categorization proposed in Kil et al. (2024), while extending it by adding two new types of differences, Viewpoint and Action. In total, VLM-SubtleBench encompasses *ten* difference types: **Attribute** captures variations in object properties such as color, size, or shape; **State** reflects the condition of an object, such as whether an apple is peeled; **Emotion** addresses comparative judgments of facial expressions; **Temporal** involves identifying which image corresponds to a later stage of an event; **Spatial** describes changes in arrangement or relative position; **Existence** refers to whether an object is missing; **Quantity** handles whether the number of objects differs across images; **Quality** captures degradations such as blur, noise, or overexposure; **Viewpoint** reflects changes in camera perspective; and **Action** denotes differences in human or animal poses or activities. Together, these ten categories establish a comprehensive taxonomy of subtle differences, spanning from low-level visual variations to high-level semantic changes.

### 3.2 DATASET CURATION

For each difference category, we curate paired images from diverse sources and generate comparative question-answer pairs through a mix of rule-based, annotation-driven, and model-assisted methods. Existing datasets with rich ground-truth labels and metadata enable systematic pairing and QA construction, while synthetic edits and primitives provide controlled settings for specific attributes. Below we briefly summarize the curation strategy for each category. Refer to appendix A for details.

**Attribute.** We use four sources: MVTEC-AD (Bergmann et al., 2019) for industrial inspection, COCO (Lin et al., 2014) for natural images, MIMIC-Diff-VQA (Hu et al., 2023) for medical domain, and synthetic primitives. In MVTEC-AD, pairs are formed by selecting anomalies of the same type but with different severity, using pixel-level defect annotations to order defect size. In COCO, we first identify images containing a single object using instance segmentation annotations, then use GPT-4o to determine the object's color and suggest a similar but distinct alternative color. We then apply the image editing model *Gemini-2.5-flash-image-preview* (also known as "nano-banana") to change the object's color accordingly, producing minimally different pairs. For the medical domain, we leverage MIMIC-Diff-VQA, which provides chest X-ray pairs from different visits of the same patient. We reformulate the original clinical questions into layperson-friendly forms using GPT-4o (e.g., "lung opacity progression" → "Which image shows whiter lungs?"). For synthetic data, we

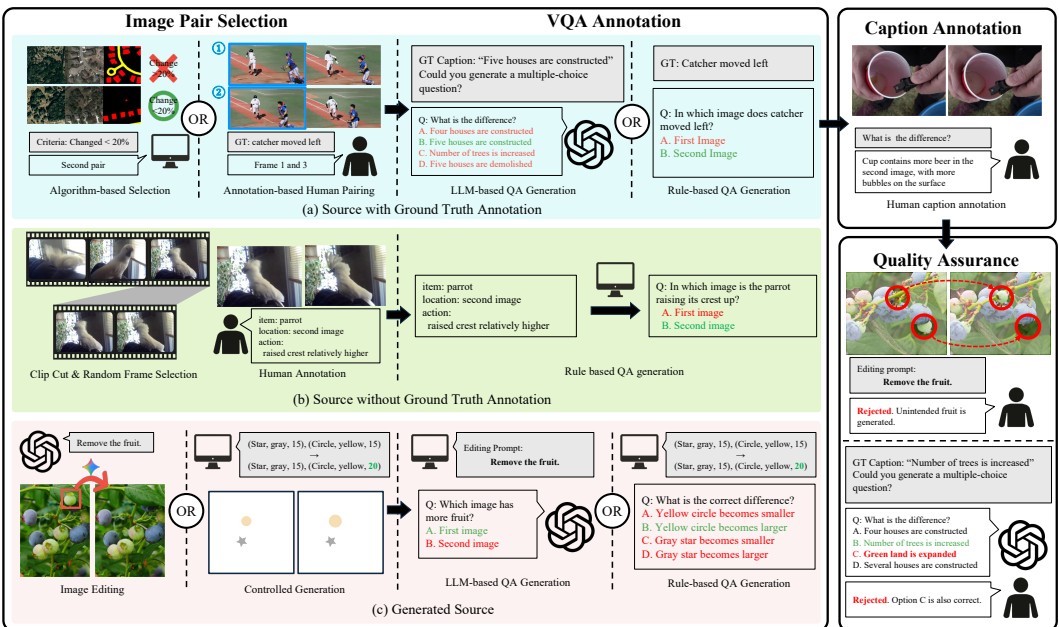

Figure 3: Data Curation Pipeline of VLM-SubtleBench.

render primitive scenes and apply subtle brightness or size modifications to a single object. QAs then ask which image has a larger defect, which color is stronger, which shape is bigger, or what change occurred.

**State.** From MVTEC-AD, we focus on object breakage or cracks. Akin to attribute, image pairs are constructed by sampling images with different levels of damage, with annotations guiding the relative severity. The resulting QAs ask which crack is larger or which state shows more breakage. We also include natural domain pairs from ChangeIt (Souček et al., 2022), where human annotators manually annotated the state-modifying action together with the object states in a set of internet videos. From each video, we sample multiple frames based on the provided state information, and human annotators then select frame pairs that capture the object before and after the state-modifying action. The QA pairs then ask which image reflects a greater degree of state modification. For example, if the object is an apple and the action is peeling, the QA would ask "In which image is the apple peeled more?".

**Emotion.** We draw images from emotional video datasets—CREMA-D, RAVDESS, AFEW-VA, and DAiSEE (Cao et al., 2014; Livingstone & Russo, 2018; Kossaifi et al., 2017; Gupta et al., 2016)—all of which provide clip-level emotion annotations. From these clips, we randomly sample frames and construct paired examples based on the relative intensity of expressed emotion. The QA pairs ask which image conveys a stronger or weaker emotion, based on the annotations.

**Temporal.** From YT8M (Abu-El-Haija et al., 2016) and VLM4D (Zhou et al., 2025) video datasets, we randomly select two frames from the same clip. Their temporal order is determined by timestamps, and QA pairs ask which image depicts the earlier event. This task requires models to capture temporal progression rather than static differences, sometimes relying on common-sense knowledge (e.g., a boat cutting through water can only move forward, not backward).

**Spatial.** We use VLM4D, which provides 4D annotations of the object's translational and rotational motions in video. From each video, we uniformly sample multiple frames. Human annotators then select frame pairs that visually align with the ground-truth motion annotations. Using these pairs and their associated motion data, we generate QA that ask the spatial changes resulting from the motion. The model is required to identify which transformation the object has undergone.

**Existence.** For aerial imagery, LEVIR-MCI (Liu et al., 2024a) provides image pairs of the same location across years, along with segmentation maps capturing object-level appearance or disappearance and human-labeled difference captions. We also construct synthetic settings by rendering

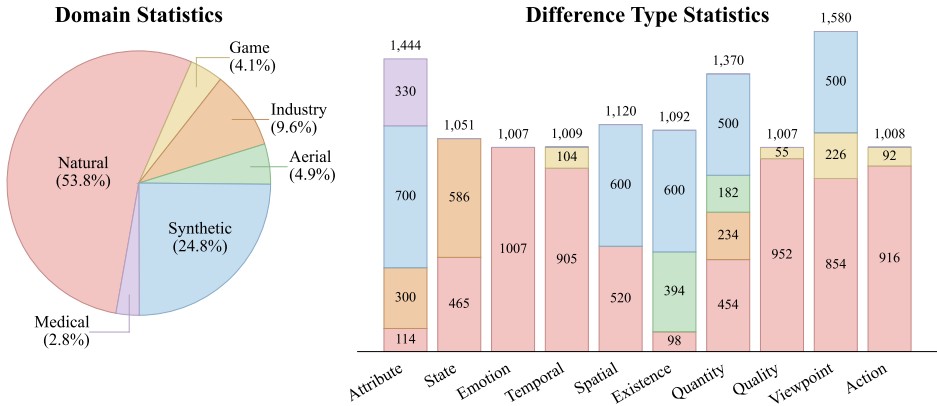

Figure 4: Statistics of the test split of VLM-SubtleBench.

primitives and removing one object to create a pair. Additionally, COCO is used to remove objects from natural images, similar to attribute. For synthetic data, we render primitive scenes containing multiple types of shapes and colors, and then add or remove one instance. In all cases, QAs ask what has disappeared or appeared between the two images, grounded in annotations or edit prompts.

**Quantity.** We combine multiple domains to capture differences in object counts. MVTEC-LOCO (Bergmann et al., 2022) provides annotated anomaly and normal images, enabling comparisons of object multiplicity. UCF-QNRF (Idrees et al., 2018) offers crowded street scenes with dense human annotations, from which we sample pairs with count differences. Aerial imagery datasets (LEVIR-MCI, UBC (Huang et al., 2022)) allow comparisons of building counts. Synthetic datasets and Gemini-edited images (MegaFruit (Wang et al., 2024)) further introduce controlled object additions. QAs consistently ask which image contains more objects, people, or buildings. For synthetic data, we render primitive scenes containing a single type of shape and color, and then create quantity differences by adding or removing one instance of the shape. The QA pairs then ask which image contains more (or fewer) shapes.

**Quality.** From YT8M, we extract ten random frames per video and present them to human annotators, who select the frame with the best visual quality and the one with the worst, based on the presence of blur, noise, overexposure, or compression artifacts. The selected pair forms a quality comparison sample. Based on these annotations, we construct binary QA pairs that ask which image has better or worse quality.

**Viewpoint.** CameraBench (Lin et al., 2025) provides camera-centric and object-centric annotations describing translations and rotations. From each video, we uniformly sample frames, and human annotators then select pairs of frames that are visually aligned with ground-truth camera annotations (e.g., sufficient visual cues for viewpoint change). Based on these pairs, we construct binary QA tasks: for example, if the camera rotated to the right, the question asks "In which direction did the camera move?" with answer choices right or left. Similarly, for object-centric annotations, QAs ask whether the camera orbited an object clockwise or counterclockwise.

In case of synthetic data, we render primitive scenes and simulate camera motion by translating the objects or by rotating them clockwise or counterclockwise around the center. From the six possible transformations, we randomly sample four options to form multiple-choice QAs, where the model must identify the ground-truth motion.

**Action.** From YT8M, we sample pairs of frames that capture changes in human or object actions and interactions. For each pair, human annotators identify a salient entity whose action differs across the two frames and provide (i) the entity description and (ii) a short free-form action phrase for each frame. We then generate QA pairs using the template "In which image is the {item} {action}?" and minimally refine grammar with GPT-4o.

### 3.3 DIFFERENCE CAPTIONS

In practical scenarios across diverse VLM application domains, the ability to directly describe differences between two images is crucial. To enable such forms of evaluation, we additionally collected human-written captions explicitly highlighting the differences between paired images, thereby complementing the comparative QA tasks. We sampled 1,200 random image pairs (10% of the test split) for human caption annotation. Annotators were instructed to identify at least one difference between the two images and to write a comparative caption using the annotation interface.

### 3.4 DATASET STATISTICS

Each difference category contains at least **1K** question–answer pairs, resulting in a total of **13K pairs**. Every difference type includes data from the natural domain, where foundation models are most commonly applied in practice. We split the dataset into a test set (11.7K) and a validation set (1.3K). The test set is used for evaluation, while the validation set is used for fine-tuning models in our experiments. Figure 4 presents the statistics of VLM-SubtleBench test set.

## 4 EXPERIMENT

### 4.1 EXPERIMENT SETUP

**Models.** We evaluate both open-source and proprietary vision–language models. For the open-source side, we use the Qwen2.5-VL (Bai et al., 2025) family at three scales (7B, 32B, and 72B), as well as LLaVA-NeXT and LLaVA-OneVision (both 7B). For proprietary models, we include GPT-4o (Achiam et al., 2023), o3, GPT-5-main, GPT-5-thinking, Claude-sonnet-4 (Anthropic, 2025), Gemini-2.5-flash (Team, 2025), and Gemini-2.5-pro. This set spans both non-reasoning models (e.g., GPT-4o, GPT-5-main) and reasoning-oriented models (e.g., o3, GPT-5-thinking).

**Prompting Strategies.** To better understand the role of prompting, we experiment with several strategies. We adopt the standard Chain-of-Thought (CoT) approach, which encourages models to generate intermediate reasoning before producing final answers (Wei et al., 2022). We further introduce a two-step reasoning setup in which the VLM generates responses in two stages. In the first step, the VLM is prompted to describe the differences between the two images that are relevant to the question; in the second stage, the two images, the question, and the output from the first step are provided together to answer the question. We also augment images with a grid and instruct the models to parse them sequentially along the horizontal axis (Izadi et al., 2025). To investigate how models handle multiple images, we test different fusion techniques: (i) horizontally concatenating the two images into a single composite input, (ii) creating an overlap image by averaging the pixel values of the two input images and using it together with the original images, (iii) generating a grayscale subtraction image by computing the absolute pixel-wise difference, normalizing it by the maximum value to highlight regions of change, and providing it along with the original images, and (iv) highlighting regions of interest by retaining pixels with large differences, clustering adjacent pixels, and drawing bounding boxes around at most three of the largest clusters to emphasize the main regions of change. Further details of these prompting techniques are provided in Appendix B.2.

**Evaluation Metric.** We use task-appropriate metrics to evaluate model performance. For multiple-choice questions, performance is measured by accuracy, capturing the proportion of correct answers. For the captioning task, we apply cosine similarity score (CSS) (Reimers & Gurevych, 2019) and LLM-as-a-judge (Zheng et al., 2023; Lin et al., 2025), to assess the quality and relevance of generated captions. See Appendix B.3 for details on the CSS and LLM-as-a-judge evaluations.

### 4.2 BENCHMARK RESULTS

**Multiple-Choice Questions.** Table 2 summarizes the performance of proprietary and open-source VLMs on VLM-SubtleBench. Among proprietary models, GPT-5-thinking achieves the strongest results overall, ranking first in 7 out of the 10 difference types and yielding the highest average accuracy. Other reasoning-oriented models such as o3 and Gemini also show strong performance, highlighting the advantage of models explicitly designed for reasoning. Within the open-source models, Qwen2.5-VL-72B stands out with competitive accuracy, in some cases approaching that of

Table 2: Performance of open-source and proprietary vision-language models in VLM-SubtleBench. Human evaluation was conducted on a randomly selected 10% of the samples.

| Model | AT | ST | EM | TM | SP | EX | QN | QL | VP | AC | AVG |
|---|---|---|---|---|---|---|---|---|---|---|---|
| Random Guess | 35.9 | 50.0 | 50.0 | 50.0 | 36.6 | 23.2 | 48.9 | 50.0 | 42.1 | 50.0 | 43.3 |
| Human Eval | 92.0 | 93.0 | 93.0 | 93.0 | 95.0 | 97.0 | 97.0 | 99.0 | 98.0 | 98.0 | 95.5 |
| *Open-source* | | | | | | | | | | | |
| LLaVA-NeXT-7B | 37.0 | 51.3 | 51.8 | 47.4 | 37.3 | 25.6 | 49.5 | 48.0 | 43.7 | 46.9 | 43.6 |
| LLaVA-OneVision-7B | 41.6 | 56.8 | 73.9 | 48.7 | 35.5 | 44.2 | 54.9 | 62.7 | 49.1 | 60.5 | 52.0 |
| Qwen2.5-VL-7B | 46.5 | 63.7 | 87.8 | 50.2 | 39.5 | 73.8 | 58.0 | 70.9 | 47.5 | 69.3 | 59.4 |
| Qwen2.5-VL-32B | 48.3 | 64.0 | 85.3 | 50.4 | 43.6 | 84.2 | 67.5 | 72.5 | 47.4 | 72.0 | 62.2 |
| Qwen2.5-VL-72B | 53.9 | 68.9 | 85.9 | 49.9 | 47.8 | 81.7 | 67.7 | 78.4 | 56.2 | 74.1 | 65.4 |
| *Proprietary* | | | | | | | | | | | |
| GPT-4o | 51.5 | 73.6 | 89.5 | 52.7 | 42.4 | 60.6 | 58.2 | 72.4 | 51.4 | 76.7 | 61.6 |
| o3 | 78.0 | 79.5 | 92.9 | **60.4** | 55.1 | 82.2 | 78.2 | **87.6** | 64.6 | 84.8 | 75.7 |
| GPT-5-main | 72.9 | 78.4 | 92.7 | 53.6 | 50.1 | 75.4 | 72.6 | 84.5 | 57.5 | 83.6 | 71.3 |
| GPT-5-thinking | **83.6** | **80.7** | **93.1** | 60.2 | **59.9** | 85.4 | **79.9** | 84.8 | **68.5** | 84.9 | **77.8** |
| Claude-sonnet-4 | 48.9 | 64.7 | 83.3 | 49.3 | 48.7 | **87.5** | 63.1 | 70.8 | 53.5 | 66.3 | 62.6 |
| Gemini-2.5-flash | 49.3 | 72.5 | 88.4 | 53.9 | 40.7 | 73.2 | 60.0 | 77.1 | 51.8 | 72.3 | 62.5 |
| Gemini-2.5-pro | 55.3 | 76.4 | 89.8 | 57.6 | 44.8 | 79.9 | 68.0 | 84.8 | 60.3 | 76.8 | 68.2 |

proprietary systems. Among the 7B-scale models, Qwen2.5-VL-7B achieved the highest accuracy, followed by LLaVA-OneVision-7B, while LLaVA-NeXT-7B showed lowest performance.

Across different difference types, VLMs show strong performance on emotion, with GPT-5-thinking achieving 93.1% accuracy. In contrast, all models perform weakly on temporal, spatial, and viewpoint differences, which require common-sense reasoning (e.g., predicting the future position of a person or distinguishing between object and camera motion) and spatial understanding. These findings underscore the need for VLMs to incorporate richer spatial–temporal representations.

**Captioning.** Table 3 presents the performance of VLMs on VLM-SubtleBench's captioning task. Similar to VQA tasks, GPT-5-thinking achieves the strongest overall performance across all metrics. However, when captions are evaluated with the LLM-as-a-judge metric, its accuracy reaches only 43.0%, leaving a noticeable gap compared to ground-truth captions. Large open-source VLMs, such as Qwen2.5-VL-32B/72B, perform on par with proprietary models in terms of CSS score, but exhibit substantially lower performance under the LLM-as-a-judge evaluation. In particular, Qwen2.5-VL-72B scores 24.3, which is significantly behind GPT-5-thinking's 43.0.

## 4.3 Effect of Prompting

Table 4 reports the effect of different prompting strategies. Adding reasoning steps before the answer improved performance in 9 out of 10 domains. While such gains are intuitive in tasks like *temporal* that require world knowledge, it is particularly interesting that reasoning also boosts performance in tasks such as *attribute* and *quality*, where success hinges on capturing fine-grained visual differences. This suggests that explicit textual reasoning supports not only abstract inference but also subtle perceptual discrimination, consistent with our main finding that models with stronger inherent reasoning achieve higher accuracy. In contrast, the two-step reasoning approach leads to a slight decrease in performance. We observe that the model frequently produces intermediate descriptions indicating "no difference" in the first stage, which results in incorrect final predictions. The highlighting method yields a modest improvement in performance. It is particularly effective on datasets with limited variations (e.g., synthetic data); however, its performance declines on datasets exhibiting substantial variations in brightness or image quality (e.g., YT8M), where bounding boxes often fail to accurately localize regions of change.

Other strategies generally led to performance drops. In particular, concatenating two images into a single input, a common setup in prior work (Kil et al., 2024; Jiao et al., 2025), degraded accuracy in 9 out of 10 domains. Overlap and subtract showed mixed effects: they yielded clear gains in spatial and existence tasks where only objects change under fixed views, and in viewpoint tasks where the scene is mostly static with camera movements. However, in other domains these strategies provided little or no benefit, reflecting their dependence on highlighting layout differences between images.

Table 3: Performance of open-source and proprietary vision-language models in VLM-SubtleBench captioning.

| Model | CSS | LLM-Judge |
|---|---|---|
| *Open-source* | | |
| Qwen2.5-VL-7B | 0.42 | 12.7 |
| Qwen2.5-VL-32B | 0.52 | 22.7 |
| Qwen2.5-VL-72B | 0.54 | 24.3 |
| *Proprietary* | | |
| GPT-4o | 0.54 | 25.2 |
| o3 | 0.56 | 38.1 |
| GPT-5-main | **0.57** | 37.2 |
| GPT-5-thinking | **0.57** | **43.0** |
| Claude-sonnet-4 | 0.54 | 24.8 |
| Gemini-2.5-flash | 0.50 | 25.9 |
| Gemini-2.5-pro | 0.52 | 29.4 |

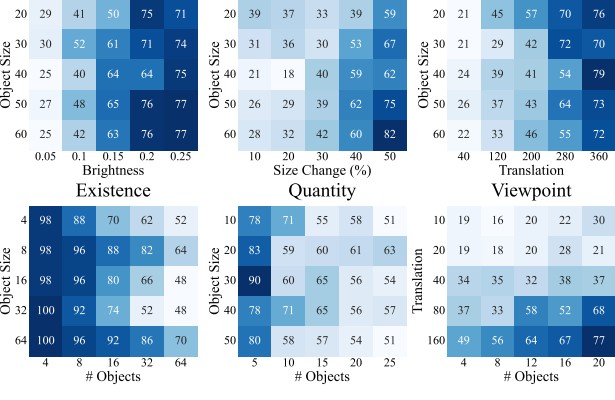

Figure 5: Performance heatmaps of GPT-4o on synthetic data under controlled difficulty factors.

Table 4: Effect of prompting strategies and fine-tuning in VLM-SubtleBench.

| Model | AT | ST | EM | TM | SP | EX | QN | QL | VP | AC | AVG |
|---|---|---|---|---|---|---|---|---|---|---|---|
| Random Guess | 35.9 | 50.0 | 50.0 | 50.0 | 36.6 | 23.2 | 48.9 | 50.0 | 42.1 | 50.0 | 43.3 |
| *Prompting Strategies* | | | | | | | | | | | |
| GPT-5-main | 72.9 | 78.4 | 92.7 | 53.6 | 50.1 | 75.4 | 72.6 | 84.5 | 57.5 | 83.6 | 71.3 |
| + Reasoning | **76.5** | **79.1** | 91.2 | 56.1 | 51.6 | 80.2 | **75.8** | **86.1** | 57.0 | **85.4** | **73.1** |
| + Two-Step Reasoning | 70.8 | **79.1** | **93.4** | **56.9** | 47.4 | 81.3 | 66.4 | 83.5 | 58.0 | 83.6 | 71.0 |
| + Grid | 71.6 | 77.5 | 89.1 | 52.8 | 51.0 | 75.8 | 72.5 | 82.6 | 57.2 | 84.2 | 70.6 |
| + Concat | 70.3 | 77.9 | 92.2 | 51.6 | 44.6 | 75.5 | 64.8 | 81.2 | 52.3 | 82.4 | 68.2 |
| + Overlap | 69.5 | 76.7 | 91.8 | 52.4 | 53.6 | 76.1 | 69.1 | 79.0 | 58.8 | 83.0 | 70.2 |
| + Subtract | 73.8 | 76.4 | 91.8 | 50.9 | **55.4** | 78.0 | 69.8 | 80.1 | **60.0** | 82.0 | 71.2 |
| + Highlight | 71.1 | 75.2 | 92.0 | 51.1 | 54.9 | **86.5** | 74.3 | 77.8 | 57.3 | 82.9 | 71.5 |
| *Fine-Tuning* | | | | | | | | | | | |
| Qwen-2.5-VL-7B | 46.5 | 63.7 | 87.8 | 50.2 | 39.5 | 73.8 | 58.0 | 70.9 | 47.5 | 69.3 | 59.4 |
| + Fine-tuned | **62.0** | **69.1** | **92.2** | **52.5** | **47.0** | **85.3** | **77.0** | **85.9** | **57.5** | **75.4** | **69.5** |

## 4.4 CONTROLLED EVALUATION WITH SYNTHETIC DATA

**Setup.** We leverage synthetic data generation to systematically manipulate task difficulty, which allows precise control over the factors that may cause VLMs to fail. For each difference type, we selected two primary factors that strongly influence difficulty and varied them along a controlled axis. Specifically, for *attribute*, we considered the size of changed objects and the magnitude of variation (brightness shifts in [0, 1] for color, or size-change ratios for scale). For *spatial*, we manipulated object size and the degree of translation. For *existence* and *quantity*, the two axes were object size and scene complexity, defined as the total number of objects. For *viewpoint*, we varied camera translation and scene complexity. For each configuration, we generated 100 paired images to probe VLM performance. Evaluation was performed using GPT-4o. Notably, for *quantity* differences, random guessing achieves 50%, while for the other categories the baseline is 25%.

**Results.** Our experiments reveal distinct failure modes across difference types. For *attribute (color)*, the decisive factor is the magnitude of brightness change: the model requires shifts of roughly 25% to show strong performance above 70%, while smaller changes, e.g., 5%, lead to random-like performance. In the *attribute (size)* condition, the model depends more on the absolute size of the changed object than on relative scaling, achieving reliable accuracy only when large objects undergo substantial transformations. For *spatial* differences, accuracy is largely influenced by both translation and object size, with the model responding more strongly to relative displacement than absolute pixel shifts; notably, smaller objects moving larger relative distances are easier to detect. In the *existence* setting, scene complexity emerges as the dominant factor: accuracy is nearly perfect with four or fewer objects but rapidly degrades to below 60% once scenes exceed 32 objects, though larger disappearing objects remain easier to track. Similarly, in *quantity*, performance remains high, nearly 80% in simple scenes with 5 objects, but drops to close to 60% in complex scene

Table 5: Rank correlation of our benchmark and MLLM-CompBench with MMAD and QAG.

| | MMAD | QAG |
|---|---|---|
| VLM-SubtleBench | **0.8424** | **0.7212** |
| MLLM-CompBench | 0.8110 | 0.7195 |

Table 6: Downstream performance after fine-tuning on each benchmark.

| | MMAD | QAG |
|---|---|---|
| Qwen2.5-VL-7B | 65.0 | 34.4 |
| + VLM-SubtleBench | **69.6** | **35.5** |
| + MLLM-CompBench | 66.3 | 32.2 |

containing ten or more objects, approaching the 50% random baseline. Finally, for *viewpoint*, the model shows an interesting opposite trend: performance improves as scene complexity increases, benefiting from richer visual cues, and stable accuracy requires camera translations of around 160 pixels (which is 27% of the image height).

## 4.5 EFFECT OF FINE-TUNING

To evaluate whether additional supervision can mitigate the comparative reasoning challenge, we fine-tune Qwen2.5-VL-7B using the validation set. Table 4 presents the results. Fine-tuning yields consistent performance improvements across all difference types, with particularly notable gains in existence, quantity, and quality categories. In contrast, the spatial and temporal dimensions showed more modest gains, suggesting that richer spatial–temporal reasoning rather than in-distribution adaptation may be required for further progress. Despite these improvements, a substantial gap remains compared to GPT-5-thinking and human performance, indicating that broader data diversity and advanced training method could offer promising future directions beyond the present scope. Details on fine-tuning are provided in Appendix B.5.

## 4.6 REAL-WORLD RELEVANCE ANALYSIS

We assess the real-world relevance of VLM-SubtleBench through correlation and transfer studies on industrial anomaly detection (MMAD (Jiang et al., 2025)) and aerial surveillance (QAG-360k (Li et al., 2024)), both requiring fine-grained visual discrimination. Table 5 reports the rank correlations (Spearman, 1904) between each benchmark and the downstream tasks. Across models, our benchmark shows higher rank correlations with MMAD and QAG-360K than MLLM-CompBench, suggesting that it better captures the comparative cues underlying downstream performance. Model-wise results are summarized in Appendix C.1.

To evaluate practical transfer, we fine-tune Qwen2.5-VL-7B on a validation split of VLM-SubtleBench and compare it with an equally sized subset of MLLM-CompBench. Table 6 reports the resulting downstream accuracies on MMAD and QAG-360K. Fine-tuning on VLM-SubtleBench yields larger gains on both application benchmarks, whereas fine-tuning on MLLM-CompBench provides limited or even negative transfer. These results indicate that the subtle, fine-grained difference types in VLM-SubtleBench more effectively encode cues for real-world perceptual reasoning.

## 5 DISCUSSION

**Conclusion.** We introduce VLM-SubtleBench, a benchmark for evaluating *subtle comparative reasoning* in vision–language models. VLM-SubtleBench focuses on image pairs with subtle changes across ten difference types and six domains, where humans succeed but current VLMs struggle significantly. Our evaluation reveals systematic model–human gaps across difference types and domains, and controlled synthetic studies expose consistent failure modes, positioning VLM-SubtleBench as both a rigorous benchmark and a diagnostic tool for future model development.

**Broader Impact.** Detecting subtle visual changes is a core requirement for VLM-based agents operating in dynamic environments. In *game* environments, where LLM agents must interpret complex visual states across diverse genres (Park et al., 2026), perceiving fine-grained changes such as character movements is crucial, and VLM-SubtleBench can serve as a diagnostic tool for such perceptual capability. In *robotic* manipulation and navigation, detecting minute changes in object pose or state is essential for reliable physical interaction. In *GUI* agent scenarios, recognizing subtle interface changes such as a button becoming enabled is critical for executing correct actions.

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

## A  BENCHMARK CURATION DETAIL

Table 7 provides a detailed breakdown of the test split by difference type, domain, and data source.

Table 7: Number of test examples per difference type, domain, and source.

| Type | Domain | Source | # |
|---|---|---|---|
| Attribute | Natural | COCO | 114 |
| | Industry | MVTEC-AD | 300 |
| | Medical | MIMIC-Diff-VQA | 330 |
| | Synthetic | Synthetic | 700 |
| State | Natural | ChangeIt | 465 |
| | Industry | MVTEC-AD | 586 |
| Emotion | Natural | CREMA-D, RAVDESS, AFEW-VA, DAiSEE | 1007 |
| Temporal | Natural | YT8M | 613 |
| | Natural | VLM4D | 292 |
| | Game | YT8M | 104 |
| Spatial | Natural | VLM4D | 520 |
| | Synthetic | Synthetic | 600 |
| Existence | Natural | COCO | 98 |
| | Aerial | LEVIR-MCI | 394 |
| | Synthetic | Synthetic | 600 |
| Quantity | Natural | MegaFruits | 330 |
| | Natural | UCF-QNRF-ECC | 66 |
| | Natural | UBC | 133 |
| | Natural | ChangeIt | 58 |
| | Aerial | LEVIR-MCI | 49 |
| | Industry | MVTEC-LOCO | 234 |
| | Synthetic | Synthetic | 500 |
| Quality | Natural | YT8M | 952 |
| | Game | YT8M | 55 |
| Viewpoint | Natural | CameraBench | 1080 |
| | Synthetic | Synthetic | 500 |
| Action | Natural | YT8M | 916 |
| | Game | YT8M | 92 |

### A.1  ATTRIBUTE

**MVTEC-AD** (Bergmann et al., 2019) data are used to construct the attribute dataset. Each defect type is categorized as either attribute (e.g., color anomaly, contamination, glue residue) or state (e.g., crack, cut, hole); defect types that cannot be meaningfully compared in terms of severity (e.g., missing components, spatial misplacements) are excluded. For each defect type, pairs are formed between two anomaly images of the same type but with different defect sizes. Defect size is measured as the percentage of defective pixels relative to the total image area, computed from pixel-level ground-truth masks. We select pairs whose defect sizes differ by at least 0.2%, with up to 15 pairs sampled per defect type. QA pairs then ask which image shows a larger or smaller defect area.

**COCO** (Lin et al., 2014) data are also used. We first identify images containing a single object using COCO instance segmentation annotations, filtering by object-to-image area ratio (0.5–10%). GPT-4o is then prompted with the bounding-box-annotated image to simultaneously determine the object's primary color, suggest a similar but distinguishable alternative (e.g., red → orange), and generate the corresponding QA pair. The image editing model Gemini-2.5-flash-image-preview

(nano-banana) applies the color change. All edited images are manually checked by human annotators, who verify that the color change matches the intended specification and that the QA pair accurately reflects the applied modification.

**MIMIC-Diff-VQA** (Hu et al., 2023), built on the MIMIC-CXR dataset (Johnson et al., 2019), provides chest X-ray pairs from different visits of the same patient along with comparative clinical questions. Since our benchmark focuses on visually perceptible differences independent of medical expertise, we reformulate the original clinical questions into layperson-friendly forms using GPT-4o (e.g., converting "lung opacity progression" into "Which image shows whiter lungs?"). Annotators then verify the consistency between the reformulated questions and the corresponding image pairs.

**Synthetic.** We render scenes containing 2–10 shapes (circles, squares, and triangles) with distinct colors on an $800\times600$ white background. One shape is randomly selected, and either a brightness adjustment or a size modification is applied. For brightness changes, we shift the lightness channel ($L$) in OKLAB color space by 5–10%, producing a subtly brighter or darker variant of the same hue. For size changes, we scale the selected shape by 15–20%. The resulting QA pairs ask which transformation occurred and to which object, with three distractors formed by combining the wrong shape, wrong color, or opposite direction of change.

## A.2 STATE

**MVTEC-AD** pairs constructed in Section A.1 are also used for state differences, since some anomalies pertain to state rather than attribute.

**ChangeIt** (Souček et al., 2022) contains annotations of state-modifying actions and resulting object states in internet videos. A subset of the videos is manually annotated, and we use this portion of the data. From each video, we sample multiple frames corresponding to either state1, action, or state2. Based on object and action, we automatically generate questions. For example, if the object is an apple and the action is peeling, the question becomes: In which image is the apple peeled more? Annotators then select frame pairs in which the state-modifying action is more evident in the second frame (Figure 6).

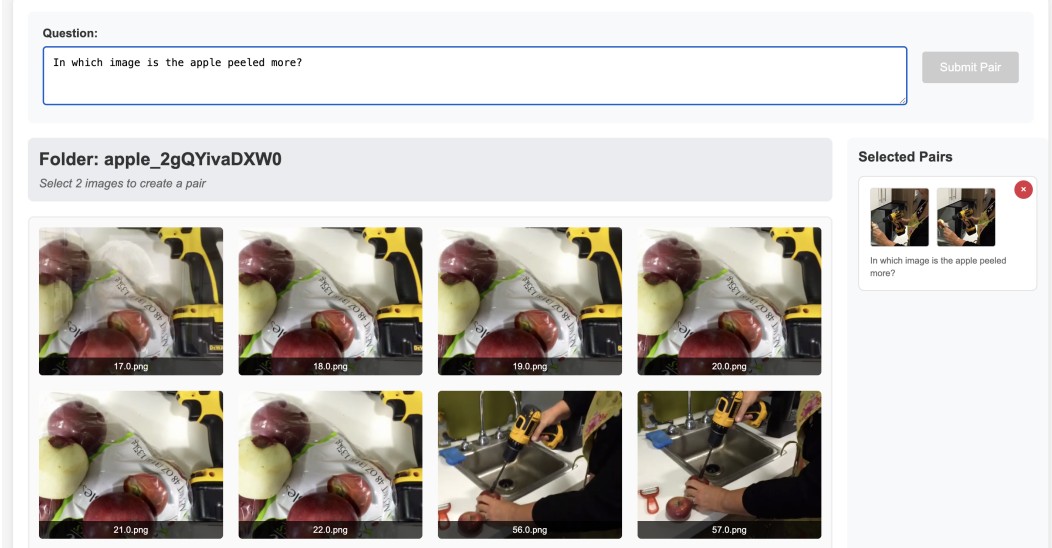

Figure 6: Example annotation interface for sources with ground-truth annotations (ChangeIt). Annotators select frame pairs from pre-sampled candidates based on the provided labels.

## A.3 EMOTION

**CREMA-D, RAVDESS, AFEW-VA, and DAiSEE** (Cao et al., 2014; Livingstone & Russo, 2018; Kossaifi et al., 2017; Gupta et al., 2016) are clip-level video datasets annotated for emotion. Specifically, CREMA-D and RAVDESS consist of actors speaking sentences with specified emotions and

intensity levels. AFEW-VA contains movie clips annotated with valence and arousal, while DAiSEE provides short video snippets capturing users' emotions. From these clips, we randomly sample frames and construct paired examples based on the relative intensity of expressed emotions. For question generation, we use the labeled emotion categories; in the case of AFEW-VA, which lacks explicit emotion labels, annotators are asked to choose the most appropriate emotion depicted in the scene.

## A.4 Temporal

**VLM4D** (Zhou et al., 2025) is a benchmark designed to evaluate the spatiotemporal reasoning capabilities of video LLMs. It provides video clips depicting exocentric and egocentric movements of people, animals, or objects, along with VQA tasks about their motion. Annotators select frame pairs that reflect the corresponding spatial change. When movement cannot be reversed (e.g., a person riding a bicycle or running), annotators label the sample as temporal. Only samples labeled as temporal are used to form the temporal pairs.

For the temporal subset, we generate questions that ask about the temporal order of the two frames. Specifically, we use two templates: "Which frame happened first?" and "Which frame happened later?"

**YT8M** (Abu-El-Haija et al., 2016) contains a diverse collection of YouTube videos spanning numerous domains. For each video, we segment clips using histogram matching and construct image pairs by randomly selecting frames with cosine similarity below 0.99, thereby avoiding frame extraction from frozen screens. Annotators then select frame pairs whose order cannot be reversed, marking them as temporal pairs (Figure 7).

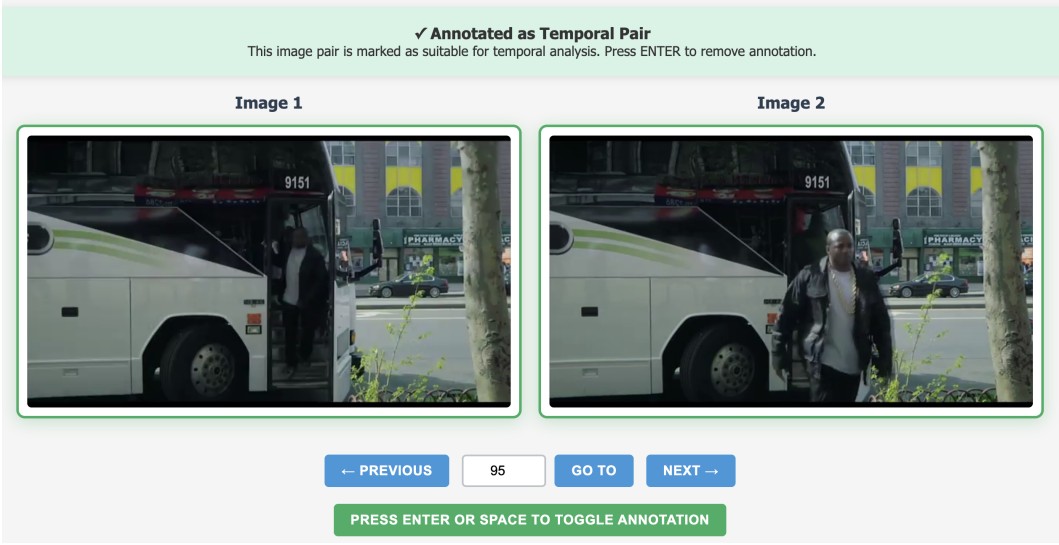

Figure 7: Example annotation interface for sources without ground-truth annotations (YT8M temporal). Annotators provide labels (e.g., whether the temporal order is irreversible) for each frame pair.

## A.5 Spatial

**VLM4D** is used to construct the spatial data. We collect frame pairs following the same procedure described in Section A.4. Samples labeled as temporal (i.e., non-reversible motions) are used to form the temporal pairs. For the spatial pairs, we use both reversible and non-reversible samples.

We then adapt the original VQA questions into a comparative form using GPT-4o. For example, if the original question is "Which direction is the horse moving towards?" with the answer "left", we reformulate it as "In which image is the horse relatively in a left position?" In addition, we

also generate a counter-directional variant by flipping the direction (e.g., "right"), in which case the correct answer becomes the earlier frame.

## A.6    EXISTENCE

**LEVIR-MCI** (Liu et al., 2024a) is an aerial dataset for image change detection. A subset of the data includes captions describing changes within the same region. We construct image pairs in which less than 20% of the total area has changed, further filtering for pairs with cosine similarity greater than 0.8. GPT-4o is then employed to paraphrase the captions in order to generate answers and distractors for the before and after images, and to determine whether each question pertains to existence or quantity.

**COCO** (Lin et al., 2014) is also used for existence pairs. We select images containing 1–10 objects with individual area ratios between 1% and 5% of the image. For each image, the largest object is chosen as the removal target. GPT-4o is prompted with the bounding-box-annotated image to generate a removal instruction and the corresponding QA pair. The image editing model Gemini-2.5-flash-image-preview (nano-banana) then removes the specified object. Human annotators verify that the removal is clean and that the QA pair accurately reflects the change.

**Synthetic.** We render scenes containing 20–30 background shapes of uniform size (35 pixels) with random types, colors, and rotations on an 800×600 white canvas. One shape is randomly selected as the target and either removed from or added to the second image, while all other shapes remain unchanged. The resulting QA pairs ask what has appeared or disappeared, with three distractors formed from unchanged background shapes paired with random appeared/disappeared labels.

## A.7    QUANTITY

**MVTEC-LOCO** (Bergmann et al., 2022) data are used to construct the quantity dataset. For each product category (e.g., juice bottle, screw bag, pushpins), we identify anomaly images with quantity-related defects (e.g., additional or missing screws, nuts, or washers) using the ground-truth defect annotations provided in the dataset's configuration. Each anomaly image is then paired with its visually closest normal image, selected by computing DINOv3 feature cosine similarity between the anomaly and all available normal images. QA pairs ask which image contains more or fewer of the specified item.

**ChangeIt** data featuring "pouring" actions are repurposed for quantity differences. Frame pair selection follows the same annotation procedure described in Section A.2. Since pouring increases the amount of liquid in a container, the resulting QA pairs ask "Which image has more {liquid} been poured?", where the liquid type varies across videos (e.g., beer, juice, milk, tea).

**LEVIR-MCI** data related to quantity are also included. We follow the same annotation method described in Section A.6.

**MegaFruits, UCF-QNRF-ECC, and UBC** (Wang et al., 2024; Idrees et al., 2018; Huang et al., 2022) are datasets containing multiple instances of fruits, people, or buildings. Annotators first draw a bounding box around a selected object in the image. A black box of the same size is then overlaid on the image, after which nano-banana is tasked with inpainting the region while ensuring that no new object is introduced. In this way, we obtain modified images in which one or more objects have been removed. The resulting QA pairs ask "Which image has {fewer, more} {object}?", where the object type depends on the source dataset (e.g., blueberries, peaches, or strawberries for MegaFruits; people for UCF-QNRF-ECC; buildings for UBC).

**Synthetic.** We render scenes containing 10–20 shapes of a single type and color (size 20–40 pixels) arranged at random non-overlapping positions on an 800×600 white canvas. To create the second image, we either add or remove one instance of the same shape type. QA pairs ask which image contains more (or fewer) of the given shape, with binary answer choices (first image / second image).

## A.8    QUALITY

**YT8M** is also used to construct the quality dataset. Many videos in YT8M contain frames of varying visual quality due to factors such as motion blur, sensor noise, overexposure, or compression

artifacts. For each video, we extract ten random frames and present them to human annotators. Annotators select two frames: one with the best visual quality and one with the worst, judging quality based on the presence or absence of blur, noise, overexposure, and artifacts. The selected pair forms a quality comparison sample. For each pair, the question is randomly chosen from one of two forms: "Which image has better quality (less blur, noise, overexposure, or artifacts)?" or "Which image has lower quality (more blur, noise, overexposure, or artifacts)?", with binary answer choices corresponding to the first or second image.

### A.9 VIEWPOINT

**CameraBench** (Lin et al., 2025) provides ground-truth camera movement annotations for video clips, covering a range of motion types such as dolly-in/out, truck-left/right, pan-left/right, tilt-up/down, pedestal-up/down, and zoom-in/out. For each video, we extract ten frames uniformly sampled from the middle 80% of the video (excluding the first and last 10%). Human annotators then select frame pairs that visually reflect the annotated camera movement, ensuring that the chosen frames provide clear visual cues for the specified motion (e.g., sufficient parallax for dolly movements or visible lateral shift for truck movements).

Based on the selected frame pairs and their associated camera movement labels, we automatically generate binary QA pairs. For each movement type, the question asks about the direction of camera motion (e.g., "In which direction does the camera move from the first image to the second image?") with two answer choices corresponding to opposite directions (e.g., left/right for truck movements, in/out for dolly movements). When a video contains multiple camera movement labels (e.g., simultaneous pan-left and tilt-up), multiple QA pairs are generated from the same frame pair, each targeting a different movement axis.

**Synthetic.** We simulate camera motion by applying a uniform transformation to all shapes in the scene on an $800 \times 600$ white canvas. For camera translation, all shapes are shifted by 20–80 pixels along a single axis, simulating a camera pan. For camera rotation, all shape positions are rotated by 5–20 degrees around the image center, and each shape's own orientation is adjusted by the same angle. QA pairs ask the direction of camera motion, with answer choices including both translation and rotation directions as distractors.

### A.10 ACTION

**YT8M** is used to construct the action dataset. We form frame pairs in the same manner as described in Section A.4. For each frame pair, annotators examine the two images and identify a salient entity whose action differs across the pair. They then provide (i) the entity description (item) and (ii) a short free-form action phrase for each image (first/second), which can capture changes in pose, activity, or interactions involving people, animals, or objects.

We generate VQA questions from these annotations. Given an annotated (*item*, *action*) from one of the two images, we create the question "*In which image is the* {*item*} {*action*}?" and minimally refine its grammar using GPT-4o.

## B EVALUATION DETAIL

### B.1 PROMPT DESIGN FOR EVALUATION TASKS

Figures 8–16 provide the exact prompts used in our experiments. Unless otherwise noted, temperature is fixed at 0.5 and the repetition penalty at 1.0.

**System Prompt**
You are a helpful assistant that answers multiple-choice questions about differences between two images.
Your task is to carefully analyze both images and identify the main difference between them.

Guidelines:
- Unless specified in the options, the difference is described in terms of the second image relative to the first.
- Respond **only** with the answer letter (A, B, C, D, etc.). Do not provide any reasoning or explanation.

**User Prompt**
Question: {question_text}

Carefully examine the images and choose the best description of the key visual difference.

Options:
{options_text}

Figure 8: *Standard prompt* used in our benchmark.

**System Prompt**
You are a helpful assistant that answers multiple-choice questions about differences between two images.
Your task is to carefully analyze both images and identify the main difference between them.

Guidelines:
- Unless specified in the options, the difference is described in terms of the second image relative to the first.
- Respond **only** in the following format. The answer should be a single letter.

### Reasoning
[explanation of the key visual difference between the two images]

### Answer
[answer (single letter)]

**User Prompt**
Question: {question_text}

Carefully examine the images and choose the best description of the key visual difference.

Options:
{options_text}

Figure 9: *Reasoning prompt* used in our benchmark.

**System Prompt (1st step)**
You are an expert image analyst. Your task is to carefully describe the differences between two images. Do not answer any questions about the images yet - only analyze and describe what is different between them.

**User Prompt (1st step)**
Please provide a careful and detailed description of the differences between these two images. Focus on what changed, what's different, or what distinguishes the first image from the second image. Using the description of the differences, you will be asked the following question and you will need to choose one correct answer. However, do not answer the question yet - just analyze the differences:

{question_text}

{options_text}

**System Prompt (2nd step)**
You are a helpful assistant that answers multiple-choice questions about differences between two images. Your task is to carefully analyze both images and identify the main difference between them. You will be provided with a description of the differences to help guide your analysis.

Guidelines:
- Use the provided difference description to understand what has changed between the images.
- Verify the described difference by examining the images.
- Unless specified in the options, the difference is described in terms of the second image relative to the first.
- Respond **only** with the answer letter (A, B, C, D, etc.). Do not provide any reasoning or explanation.

**User Prompt (2nd step)**
I am showing you two images (first and second).

Description of the differences: {diff_description}

Question: {question_text}

Based on the images and the description of differences, choose the best answer.

Options:
{options_text}

Figure 10: *Two-step reasoning prompt* used in our benchmark.

**System Prompt**
You are a helpful assistant that answers multiple-choice questions about differences between two images. The grid lines are added to both images to help you compare the objects better.
Your task is to carefully analyze both images and identify the main difference between them.

Guidelines:
- Unless specified in the options, the difference is described in terms of the second image relative to the first.
- Respond **only** with the answer letter (A, B, C, D, etc.). Do not provide any reasoning or explanation.

**User Prompt**
Question: {question_text}

Carefully examine the images and choose the best description of the key visual difference.

Options:
{options_text}

Figure 11: *Grid prompt* used in our benchmark.

**System Prompt**

You are a helpful assistant that answers multiple-choice questions about differences between two images that are concatenated horizontally (first image on the left and second image on the right, separated by a black line). Your task is to carefully analyze both images and identify the main difference between them.

Guidelines:
- Unless specified in the options, the difference is described in terms of the second image relative to the first.
- Respond **only** with the answer letter (A, B, C, D, etc.). Do not provide any reasoning or explanation.

**User Prompt**

Question: {question_text}

Carefully examine the images and choose the best description of the key visual difference.

Options:
{options_text}

Figure 12: *Concatenating prompt* used in our benchmark.

**System Prompt**

You are a helpful assistant that answers multiple-choice questions about differences between two images. Your task is to carefully analyze the images and identify the main difference between them. I am showing you four images:
1. Original first image
2. Original second image
3. Highlighted first image (with areas of significant change marked with green boxes, and other areas dimmed)
4. Highlighted second image (with the same areas marked)

The highlighted images help you focus on the most significant differences between the two images. Use them to quickly identify where the changes occur, then examine those areas carefully in the original images.

Guidelines:
- Unless specified in the options, the difference is described in terms of the second image relative to the first.
- Focus on the green-boxed regions in the highlighted images to identify where changes occur.
- Respond **only** with the answer letter (A, B, C, D, etc.). Do not provide any reasoning or explanation.

**User Prompt**

I am showing you four images:
1. Original first image
2. Original second image
3. Highlighted first image (green boxes mark significant change areas, other areas dimmed)
4. Highlighted second image (same areas marked)

The highlighted images (3 and 4) show you WHERE the main differences are located. The green boxes indicate the top 2-3 most significant change regions. Use these to guide your attention, then carefully examine those specific areas in the original images (1 and 2) to determine WHAT the difference is.

Question: {question_text}

Carefully examine the images and choose the best description of the key visual difference.

Options:
{options_text}

Figure 13: *Highlighting prompt* used in our benchmark.

**System Prompt**
You are a helpful assistant that answers multiple-choice questions about differences between two images. Your task is to carefully analyze first and second images and identify the main difference between them. The third image is the overlay of the first and second images. You may use the third image to help you analyze the difference between the first and second images.

Guidelines:
- Unless specified in the options, the difference is described in terms of the second image relative to the first.
- Respond **only** with the answer letter (A, B, C, D, etc.). Do not provide any reasoning or explanation.

**User Prompt**
I am showing you three images:
1. First image
2. Second image
3. Overlapped image (50/50 blend of first and second images)

Question: {question_text}

Carefully examine the images and choose the best description of the key visual difference of first and second images.

Options:
{options_text}

Figure 14: *Overlapping prompt* used in our benchmark.

**System Prompt**
You are a helpful assistant that answers multiple-choice questions about differences between two images. Your task is to carefully analyze first and second images and identify the main difference between them. The third image is a black-and-white difference map between the first and second images, where brighter areas indicate larger differences. You may use the third image to help you analyze the difference between the first and second images.

Guidelines:
- Unless specified in the options, the difference is described in terms of the second image relative to the first.
- Respond **only** with the answer letter (A, B, C, D, etc.). Do not provide any reasoning or explanation.

**User Prompt**
I am showing you three images:
1. First image
2. Second image
3. Black-and-white difference map between the first and second images

Question: {question_text}

Carefully examine the images and choose the best description of the key visual difference of first and second images.

Options:
{options_text}

Figure 15: *Subtracting prompt* used in our benchmark.

---

**System Prompt**

You are an expert at visual analysis and image comparison. Compare the two images and briefly describe what's different between them.

Keep your response concise and direct. Use simple phrases like "In the first image, X, however in the second image, Y" or "X appeared in the second image" or "In the first image, X is relatively Y". Avoid detailed explanations or structured lists.

**User Prompt**

Describe the differences between the two images

---

Figure 16: *Captioning prompt* used in our benchmark.

### B.2 Visual Input Construction and Examples

We describe how visual inputs are constructed for prompting, including grid overlays, image concatenation, image overlap, and image subtraction.

**Grid Overlays.** A 4×4 grid overlay is generated by drawing black lines with 30% opacity and a line width of 3 pixels on both images.

**Concatenated Images.** Image pairs are concatenated horizontally with a 1-pixel-wide black separator, forming a single composite image that is then used as the VQA input.

**Overlap Images.** Two aligned inputs are blended with equal weights (50% contribution each) in pixel space, producing a composite image that visually merges both inputs. The generated overlap image is provided to the VLM together with the original input pair (Figure 17).

**Subtraction Images.** Subtraction images are generated by computing the absolute pixel-wise difference between the aligned inputs, converting the result to grayscale, and normalizing it to highlight regions of maximal change. Specifically, for two images $I_1, I_2 \in [0, 255]^{H \times W \times 3}$,

$$G(x,y) = \frac{1}{3} \sum_{c=1}^{3} \big| I_1(x,y,c) - I_2(x,y,c) \big|,$$

$$S(x,y) = 255 \cdot \frac{G(x,y)}{\max_{u,v} G(u,v)},$$

where $S$ denotes the grayscale subtraction image. For both overlap and subtraction variants, the generated images are provided to the VLM alongside the original inputs (Figure 18).

**Highlight Images.** Highlight images are generated by drawing bounding boxes on the largest regions of change. Given the pixel-wise difference map $G$, we obtain a mask of considerable change by computing a percentile threshold: letting $\tau_p$ be the $p$-th percentile of values in $G$ (we use $p = 90$), we define the binary mask

$$M(x,y) = \begin{cases} 1, & G(x,y) > \tau_p, \\ 0, & \text{otherwise.} \end{cases}$$

Morphological closing and opening are applied to $M$ to connect nearby changes and remove noise, producing a cleaned mask with connected components $\{C_k\}$. The clusters $\{C_k\}$ are sorted by area, and the three largest clusters are selected. To avoid highlighting insignificant regions, we retain the second and third clusters only if their areas are at least 50% of the area of the largest cluster; otherwise, they are discarded. Highlight images are then constructed by dimming the background (using $\alpha = 0.5$) while preserving the original appearance only inside the selected bounding boxes. Finally, we draw a green-colored boundary around each box to emphasize the regions of change (Figure 19).

### B.3 Caption Evaluation Metrics

**Cosine similarity score.** We measure caption-level semantic agreement using cosine similarity score (CSS) (Reimers & Gurevych, 2019), implemented as cosine similarity between Sentence-

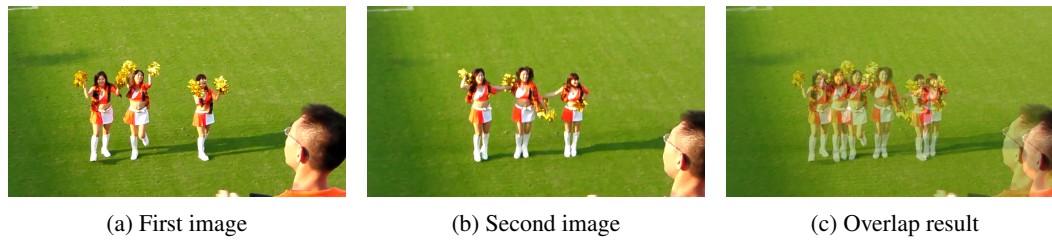

|                 |                  |                    |
| (a) First image | (b) Second image | (c) Overlap result |

Figure 17: Example of overlap-image construction. Two aligned inputs are blended with equal weights (50% each) to produce a composite image that visually merges both inputs.

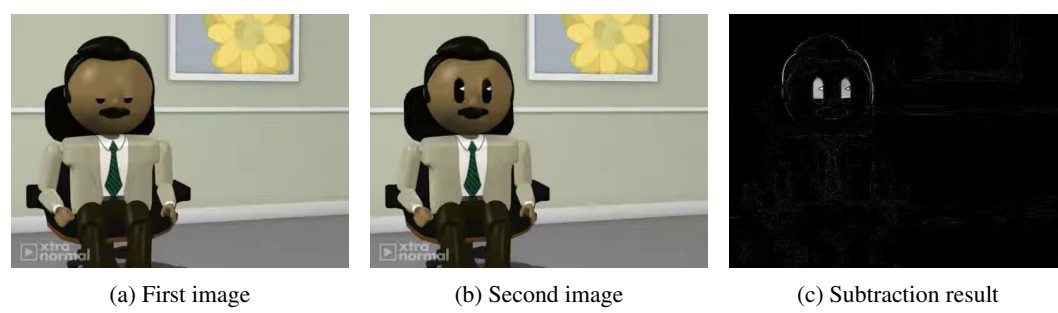

|                 |                  |                        |
| (a) First image | (b) Second image | (c) Subtraction result |

Figure 18: Example of subtraction-image construction. Given two aligned images ($I_1$, $I_2$), the subtraction map $S$ highlights regions with maximal pixel-wise change.

BERT embeddings of the ground-truth caption and the model-generated caption. We compute CSS using a SentenceTransformer model (`paraphrase-MiniLM-L6-v2`).

**LLM-as-a-judge.** Following (Lin et al., 2025), we use GPT-4o as an evaluator to judge whether the candidate caption matches the reference caption. We prompt GPT-4o with: *Reference caption: "{reference}" Candidate caption: "{candidate}" Does the candidate caption match the reference caption? Answer Yes or No.* We treat the ground-truth caption as the reference and the model-generated caption as the candidate, and compute accuracy as the fraction of outputs labeled "Yes".

### B.4 HUMAN EVALUATION

Human evaluation is conducted by sampling 1,000 examples in total: 100 examples per difference type. Since each difference type draws from multiple data sources, we first determine the number of samples per source proportionally to its contribution within the type, then randomly sample that many examples from each source. During evaluation, annotators are asked to answer questions using an interface that displays two images side by side for comparison, with the option to view images sequentially.

### B.5 FINE-TUNING SETUP

Training is performed on 4 NVIDIA A100 GPUs, each equipped with 80GB memory. We use a learning rate of 1e-5, a per-device batch size of 8, resulting in an effective batch size of 32. The models are trained for 3 epochs, with all parameters including vision encoder, projector, and language model jointly optimized.

For the transferability study (Table 6), we fine-tune Qwen-2.5-VL-7B on our 1,277-sample validation split of SubtleBench and a size-matched subset of MLLM-CompBench for comparison. For MMAD evaluation, all training instances are converted to the standardized MMAD task format, and for QAG-360K, all training samples are adapted to our standard prompt template. Aside from this dataset-specific formatting, the fine-tuning procedure is identical across all settings, and we use the same hyperparameters described above.

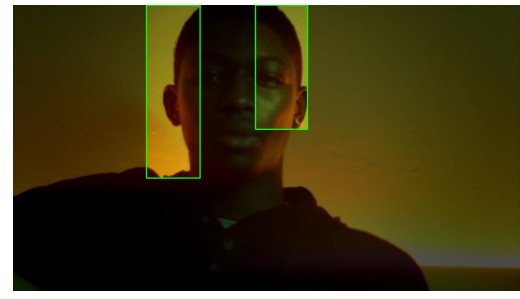 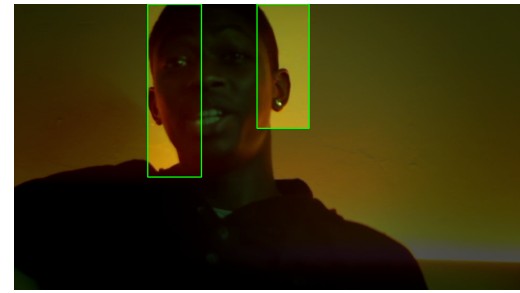

(a) Highlighted first image                    (b) Highlighted second image

Figure 19: Example highlight images. Significant regions of change, extracted via percentile thresholding and morphological filtering, are emphasized by dimming the background and drawing green bounding-box boundaries.

Table 8: Model-wise accuracy on VLM-SubtleBench, MLLM-CompBench, MMAD, and QAG-360K.

| Models | VLM-SubtleBench | MLLM-CompBench | MMAD | QAG-360K |
|---|---|---|---|---|
| Qwen-2.5-VL-7B | 59.4 | 73.6 | 65.0 | 34.4 |
| Qwen-2.5-VL-32B | 62.2 | 74.6 | 67.6 | 35.5 |
| Qwen-2.5-VL-72B | 65.4 | 76.9 | 68.9 | 41.2 |
| GPT-4o | 61.6 | 75.7 | 67.7 | 35.2 |
| o3 | 75.7 | 86.3 | 72.9 | 39.7 |
| GPT-5-main | 71.3 | 83.9 | 70.6 | 36.3 |
| GPT-5-thinking | 77.8 | 86.3 | 73.5 | 42.1 |
| Claude-sonnet-4 | 62.6 | 73.6 | 70.9 | 30.7 |
| Gemini-2.5-flash | 62.5 | 85.2 | 71.4 | 36.8 |
| Gemini-2.5-pro | 68.2 | 87.2 | 72.2 | 36.3 |

## C    ADDITIONAL RESULTS

### C.1    CORRELATION ANALYSIS WITH DOWNSTREAM BENCHMARKS

Table 8 shows the accuracy of all evaluated models across VLM-SubtleBench, MLLM-CompBench, MMAD, and QAG-360K. For MMAD evaluation, we randomly sample 10% of the test set and evaluate models using this subset. For QAG-360K evaluation, we exclude the change-ratio category due to its continuous answer format and convert the remaining question types into multiple-choice queries following our base prompt template. We then randomly sample 723 validation examples for evaluation.

### C.2    ADDITIONAL PROMPTING STRATEGY

We explore an additional prompting strategy based on pure language-only comparison.

For the *emotion*, *existence*, and *quality* categories—similar to MLLM-CompBench (Kil et al., 2024)—we ask the following questions for each image:

- **Emotion:** "Describe the emotion expressed in the image in detail and rate its intensity on a scale of 1–10."

- **Existence:** "Carefully list all objects visible in the image, including their approximate locations."

- **Quality:** "Analyze the quality of the image and rate it on a scale of 1–10, considering blur, noise, overexposure, compression artifacts, and other quality issues."

Table 9: Effect of pure language-based comparison for emotion, existence, and quality categories.

| Category | GPT-5-main | Two-stage Reasoning |
|---|---|---|
| Emotion | 92.7 | 87.8 |
| Existence | 75.4 | 63.2 |
| Quality | 84.5 | 84.6 |

Table 10: Domain-wise accuracy (%) of open-source and proprietary VLMs.

| Model | Natural | Game | Industry | Aerial | Synthetic | Medical |
|---|---|---|---|---|---|---|
| Random Guess | 49.2 | 50.0 | 50.0 | 26.9 | 29.3 | 50.0 |
| *Open-source* | | | | | | |
| LLaVA-NeXT-7B | 48.6 | 47.9 | 53.8 | 35.4 | 30.0 | 41.5 |
| LLaVA-OneVision-7B | 59.3 | 56.0 | 54.7 | 62.2 | 32.4 | 50.9 |
| Qwen2.5-VL-7B | 65.2 | 61.8 | 66.3 | 62.8 | 43.8 | 50.3 |
| Qwen2.5-VL-32B | 66.1 | 63.4 | 64.4 | 64.6 | 53.0 | 54.5 |
| Qwen2.5-VL-72B | 69.6 | 65.1 | 69.6 | 64.1 | 54.8 | 65.2 |
| *Proprietary* | | | | | | |
| GPT-4o | 68.4 | 65.8 | 71.5 | 46.9 | 45.0 | 62.4 |
| o3 | **77.3** | **76.4** | 79.3 | 71.4 | 72.5 | 68.5 |
| GPT-5-main | 74.1 | 72.1 | 77.9 | 60.8 | 63.6 | 78.8 |
| GPT-5-thinking | 77.2 | 75.5 | **81.2** | 70.7 | **78.8** | **82.4** |
| Claude-sonnet-4 | 64.2 | 60.3 | 66.3 | 74.1 | 56.3 | 54.8 |
| Gemini-2.5-flash | 68.1 | 66.5 | 73.7 | 73.4 | 43.3 | 62.4 |
| Gemini-2.5-pro | 73.2 | 71.8 | 79.7 | **75.7** | 50.6 | 68.8 |

We then perform VQA using only the generated descriptions, without providing the original images, to evaluate the model's ability to answer purely from language-based reasoning.

**Results.** The results for the pure language-based comparison are shown in Table 9. For the *emotion* and *existence* categories, this method exhibits lower performance, consistent with the findings reported in MLLM-CompBench. Interestingly, performance on the *quality* category remains nearly identical, suggesting that explicit 1–10 rating prompts may serve as a reliable intermediate representation for comparative judgment in this aspect.

### C.3 DOMAIN-WISE PERFORMANCE ANALYSIS

Table 10 shows domain-wise accuracy for proprietary VLMs. Among proprietary models, o3 and GPT-5-thinking achieve the highest scores in most categories. In particular, they exhibit a substantial performance gap over other models in the synthetic and medical domains, while both show comparatively weaker results on the aerial domain.

### C.4 EXTENDED COLOR-SENSITIVITY ANALYSIS

Inspired by (Hyeon-Woo et al., 2024), we extend our synthetic-control analysis to include a color-sensitivity axis, probing whether hue-level perceptual biases compound subtle comparative reasoning failures. We incorporate five representative colors (two green tones and three non-green: blue, red, and magenta) and systematically vary $\Delta E$ (hue shift in OKLAB space), brightness ($L$ channel), size, count, and translation. OKLAB is chosen for its perceptual uniformity, where $L$ encodes lightness and $(a, b)$ correspond to green–magenta and blue–yellow opponent axes. To isolate hue effects, $\Delta E$ adjustments are applied by modifying $(a, b)$ while keeping $L$ constant, and brightness variation fixes hue while sampling $L \in [0.4, 0.8]$. All other setup conditions follow those in Figure 5.

Results are presented in Figure 20. Consistent with prior work (Hyeon-Woo et al., 2024), models exhibit pronounced difficulty in distinguishing green hues, showing significantly lower accuracy than

for red or blue tones. Magenta elicits the most severe degradation (approaching 0%), revealing a systematic color-specific weakness in GPT-4o. In contrast, brightness variation produces no notable color-dependent gap, suggesting that VLMs are relatively invariant to luminance when performing comparative reasoning. Cross-factor analyses (color × size/count/viewpoint) show minimal interaction, implying that these tasks are largely unaffected by color sensitivity, as hue discrimination itself contributes little to the reasoning objective.

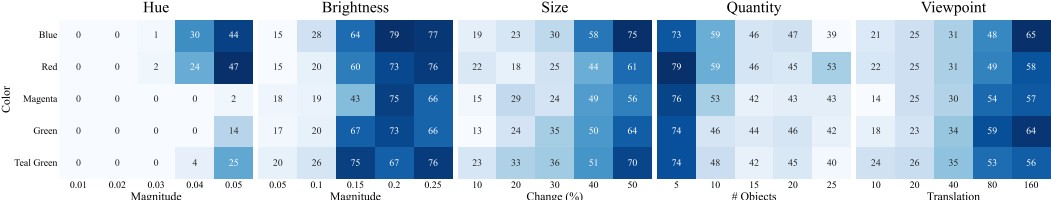

Figure 20: Color-sensitivity analysis of GPT-4o under controlled synthetic settings.

## C.5 SOURCE VALIDATION AND DISTRIBUTIONAL CONSISTENCY

To examine whether the use of nano-banana may introduce stylistic artifacts or distribution shifts that could confound evaluation, we conduct an analysis on our caption dataset. Specifically, we compare approximately 1K original (non-edited) caption pairs with subsets in which one image of each pair was reconstructed using nano-banana. Evaluation is performed on both VQA accuracy and captioning metrics (CSS and LLM-Judge). Table 11 reports the comparison between edited and non-edited pairs. Across all metrics, the differences are negligible, indicating that the use of nano-banana editing does not introduce measurable stylistic or distributional bias that could affect evaluation.

Table 11: Comparison between real and *nano-banana*-reconstructed image pairs on VQA and captioning metrics.

| Type | Accuracy | CSS | LLM-Judge |
|---|---|---|---|
| Real | 60.6 | 0.51 | 26.3 |
| Reconstructed | 60.6 | 0.51 | 27.3 |

## C.6 EFFECT OF PROMPTING IN OPEN-SOURCE VISION-LANGUAGE MODELS

Table 12 reports the effect of prompting strategies on an open-source model (Qwen2.5-VL-72B), complementing the proprietary model analysis (GPT-5-main) in the main text. Overall, we observe similar trends: reasoning slightly improves performance, while concatenation and overlap tend to degrade accuracy. Notably, the subtract strategy yields the strongest gains on spatial tasks, consistent with the proprietary model results.

Table 12: Effect of different prompting strategies in VLM-SubtleBench.

| Model | AT | ST | EM | TM | SP | EX | QN | QL | VP | AC |
|---|---|---|---|---|---|---|---|---|---|---|
| Random Guess | 35.9 | 50.0 | 50.0 | 50.0 | 36.6 | 23.2 | 48.9 | 50.0 | 42.1 | 50.0 |
| Qwen2.5-VL-72B | 53.9 | **68.9** | 85.9 | 49.9 | 47.8 | 81.7 | 67.7 | **78.4** | **56.2** | 74.1 |
| + Reasoning | 50.6 | 68.0 | **87.5** | **51.0** | 44.4 | 81.7 | 65.0 | 77.1 | 54.7 | **75.0** |
| + Grid | 54.9 | 67.6 | 85.6 | 49.8 | 46.3 | **83.4** | **70.1** | 75.7 | 54.4 | 71.1 |
| + Concat | 55.7 | 68.0 | 86.4 | 50.0 | 39.6 | 67.5 | 61.0 | 65.4 | 47.0 | 73.7 |
| + Overlap | 46.3 | 63.2 | 85.3 | 49.7 | 48.8 | 78.7 | 63.1 | 72.5 | 49.7 | 71.6 |
| + Subtract | **55.7** | 64.4 | 85.6 | 49.5 | **51.1** | 79.8 | 63.4 | 63.0 | 49.9 | 71.2 |

Table 13: Performance of proprietary models by data source

| Dataset | Random | GPT-4o | GPT-5-main | GPT-5-thinking | GPT-o3 | Claude-sonnet-4 | Gemini-2.5-flash | Gemini-2.5-pro |
|---|---|---|---|---|---|---|---|---|
| *Attribute* | | | | | | | | |
| MVTEC-AD | 50.0 | 76.7 | 79.0 | 82.0 | 79.0 | 66.3 | 76.3 | 83.3 |
| COCO | 25.0 | 78.1 | 82.5 | 86.0 | 84.2 | 71.1 | 69.3 | 77.2 |
| MIMIC-DIFF-VQA | 50.0 | 62.4 | 78.8 | 82.4 | 68.5 | 54.8 | 62.4 | 68.8 |
| Synthetic | 25.0 | 31.3 | 66.0 | 84.4 | 81.1 | 35.0 | 28.3 | 33.4 |
| *State* | | | | | | | | |
| MVTEC-AD | 50.0 | 67.2 | 73.5 | 75.9 | 74.9 | 64.0 | 71.5 | 75.1 |
| ChangeIt | 50.0 | 81.7 | 84.5 | 86.7 | 85.4 | 65.6 | 73.8 | 78.1 |
| *Emotion* | | | | | | | | |
| CREMA-D, RAVDESS, AFEW-VA, DAiSE | 50.0 | 89.5 | 92.7 | 93.1 | 92.9 | 83.3 | 88.4 | 89.8 |
| *Temporal* | | | | | | | | |
| YT8M | 50.0 | 52.7 | 55.4 | 62.3 | 63.0 | 49.4 | 55.8 | 60.4 |
| VLM4D | 50.0 | 52.7 | 49.3 | 54.8 | 53.8 | 49.0 | 49.3 | 50.7 |
| *Spatial* | | | | | | | | |
| VLM4D | 50.0 | 58.5 | 57.3 | 62.5 | 60.4 | 52.1 | 56.5 | 58.5 |
| Synthetic | 25.0 | 28.5 | 43.8 | 57.7 | 50.5 | 45.7 | 27.0 | 33.0 |
| *Existence* | | | | | | | | |
| LEVIR-MCI | 20.0 | 43.7 | 59.1 | 74.6 | 73.6 | 79.2 | 80.5 | 79.7 |
| COCO | 25.0 | 77.6 | 86.7 | 80.6 | 88.8 | 83.7 | 75.5 | 92.9 |
| Synthetic | 25.0 | 69.0 | 84.2 | 93.3 | 86.8 | 93.5 | 68.0 | 77.8 |
| *Quantity* | | | | | | | | |
| MVTEC-LOCO | 50.0 | 75.6 | 87.6 | 93.2 | 90.6 | 71.8 | 75.6 | 86.8 |
| MegaFruits | 50.0 | 50.9 | 59.7 | 66.1 | 67.0 | 57.9 | 51.5 | 62.7 |
| UCF-QNRF-ECC | 50.0 | 51.5 | 71.2 | 78.8 | 83.3 | 50.0 | 56.1 | 71.2 |
| UBC | 50.0 | 51.9 | 60.9 | 59.4 | 66.9 | 61.7 | 52.6 | 59.4 |
| LEVIR-MCI | 20.0 | 59.2 | 73.5 | 69.4 | 65.3 | 67.3 | 73.5 | 87.8 |
| ChangeIt | 50.0 | 41.4 | 58.6 | 55.2 | 56.9 | 50.0 | 60.3 | 62.1 |
| Synthetic | 50.0 | 59.4 | 79.0 | 92.4 | 86.0 | 65.8 | 59.4 | 63.2 |
| *Quality* | | | | | | | | |
| YT8M | 50.0 | 72.4 | 84.5 | 84.8 | 87.6 | 70.8 | 77.1 | 84.8 |
| *Viewpoint* | | | | | | | | |
| CameraBench | 50.0 | 56.2 | 63.8 | 69.9 | 68.5 | 57.6 | 58.1 | 64.8 |
| Synthetic | 25.0 | 41.0 | 44.0 | 65.4 | 56.2 | 44.8 | 38.2 | 50.4 |
| *Action* | | | | | | | | |
| YT8M | 50.0 | 76.7 | 83.6 | 84.9 | 84.8 | 66.3 | 72.3 | 76.8 |

## C.7 PERFORMANCE OF PROPRIETARY MODELS BY DATA SOURCE

Table 13 shows the performance of proprietary models by data source. Across models, o3 and GPT-5-thinking achieved the highest scores in most categories, exhibiting a substantial performance gap over other models in the synthetic and medical domains, while both showed comparatively weaker results on the aerial domain. Across domains, models generally performed strongest on natural and industrial imagery, while performance degraded in synthetic and aerial settings.

