# OpenReview forum: "VLM-SubtleBench: How Far Are VLMs from Human-Level Subtle Comparative Reasoning?"
_ICLR.cc/2026/Conference — ICLR 2026 Poster_

### Official Review · Reviewer_WLtr · 2025-10-28

**Soundness:** 3
**Presentation:** 3
**Contribution:** 3
**Rating:** 8
**Confidence:** 4

**Summary:**

The paper introduces a new benchmark designed to evaluate whether multimodal LLMs can distinguish subtle differences between similar images (i.e., perform subtle comparative reasoning). The benchmark is comprehensive: it contains 11.7K triplets of image pairs, questions, and answers, covering ten types of differences (e.g., attribute or state) across five image domains, including natural, industrial, and medical. The authors evaluate multiple recent MLLMs, both open-source and proprietary, on this benchmark and show that subtle comparative reasoning remains a significant challenge for current models.

**Strengths:**

- The benchmark is comprehensive, encompassing a large number of data points as well as diverse visual types and domains.

- The paper identifies the remaining limitations of recent MLLMs, providing valuable insights into where the research community should focus to further improve these models.

- The paper is well-written and easy to follow.

**Weaknesses:**

- A more detailed analysis is needed. e.g., why do MLLMs struggle more with certain types of comparisons? Why do some models perform better than others?

- The paper currently reports results but lacks discussion or suggestions on how to improve subtle comparative reasoning in MLLMs.

**Questions:**

- Similar to the study (Table 3) in MLLM-CompBench, what would happen if the models were first asked to analyze two images separately and then compare them using a purely language-based question, instead of being given both images simultaneously?

- Did annotators label all the data? How many annotators were involved in building this benchmark, and what were their backgrounds?

- It seems that only a test set is provided. Do you also have training or validation sets? It would be interesting to see whether fine-tuning on such data could improve performance.

---

> ### Author Response · Authors · 2025-11-21
> **Official Comment by Authors (1/2)**
>
> We sincerely appreciate the reviewer's helpful comments and positive feedback on our manuscript.
>
> `W1. A more detailed analysis is needed. e.g., why do MLLMs struggle more with certain types of comparisons? Why do some models perform better than others?`
>
> We thank the reviewer for the thoughtful question.
>
> > Why do MLLMs struggle more with certain types of comparisons?
>
> We observed that MLLMs perform notably worse on Temporal, Spatial, and Viewpoint categories. This aligns with findings from prior works [1,2,3] on single-image and video reasoning, where multimodal models often struggle with tasks involving spatio-temporal dynamic or geometric reasoning.
>
> > Why do some models perform better than others?
>
> From our experiments on proprietary models and prompting strategies, we found that explicit reasoning substantially improves model performance. Reasoning-oriented models such as o3 and GPT-5-Thinking outperformed others across most difference types. Moreover, in our prompting strategy, enabling non-reasoning models to perform reasoning before answering significantly enhanced their performance. We found that reasoning helps models better discriminate the relevant differences and interpret changes in relation to their context, leading to stronger comparative judgment. For instance, when both an object’s spatial position and the camera position changed simultaneously, reasoning models choosed correct answer more often by analyzing the object’s position relative to other surrounding objects.
>
> We will include this discussion in the revised manuscript.
>
>
> `W2. The paper currently reports results but lacks discussion or suggestions on how to improve subtle comparative reasoning in MLLMs.` & `Q3. It seems that only a test set is provided. Do you also have training or validation sets? It would be interesting to see whether fine-tuning on such data could improve performance.`
>
> We thank the reviewer for this helpful suggestion. Following the exact pipeline used to construct our test set, we additionally curated a train set comprising 10% of the original test-set size (~1k samples). We used this subset to examine whether fine-tuning on VLM-SubtleBench data can enhance performance.
>
> When fine-tuning Qwen-2.5-VL-7B on this training set, we observed consistent performance gains across all difference types. Results are summarized below:
>
> | Difference Type | Qwen-2.5-VL-7B | Qwen-2.5-VL-7B (fine-tuned) | GPT-5-thinking | Human Eval
> | --------------- | -------------- | --------------------------- |----|----|
> | Attribute       | 46.5           | 62.0                    |83.6|92.0
> | State           | 63.7           | 69.1                    |80.7|93.0
> | Emotion         | 87.8           | 92.2                    |93.1|93.0
> | Temporal        | 50.2           | 52.5                    |60.2|93.0
> | Spatial         | 39.5           | 47.0                    |59.9|95.0
> | Existence       | 73.8           | 85.3                    |85.4|97.0
> | Quantity        | 58.0           | 77.0                    |79.9|97.0
> | Quality         | 70.9           | 85.9                    |84.8|99.0
> | Camera          | 47.5           | 57.5                    |68.5|98.0
> | Action          | 69.3           | 75.4                    |84.9|98.0
>
> Although fine-tuning yields consistent performance gains, the improvements in certain difference types (e.g., temporal and spatial) are relatively modest, and a substantial gap remains compared to proprietary models and human-level performance. These findings highlight that collecting diverse high-quality comparative data is crucial for improving VLM performance in subtle reasoning settings, suggesting promising directions for future dataset construction and training strategies.
>
> **Note**. We identified an issue in the originally reported results for Qwen-2.5-VL-7B and have corrected them accordingly. During evaluation via an external API service (OpenRouter), we found that the 7B model frequently returned empty responses without any error messages, which affected the initially submitted results. All numbers reported here reflect the corrected evaluation.

---

> ### Author Response · Authors · 2025-11-21
> **Official Comment by Authors (2/2)**
>
> `Q1. Similar to the study (Table 3) in MLLM-CompBench, what would happen if the models were first asked to analyze two images separately and then compare them using a purely language-based question, instead of being given both images simultaneously?`
>
> We thank the reviewer for the insightful question and suggestion regarding two-stage reasoning. We applied the two-stage reasoning methodology to the categories where it is directly applicable: existence, emotion, and quality.
>
> We asked the following question for the description of each images:
> - Emotion: "Describe the emotion expressed in the image in detail and rate its intensity on a scale of 1-10."
> - Existence: "Carefully list all objects visible in the image, including their approximate locations."
> - Quality: "Analyze the quality of the image and rate it on a scale of 1-10, considering blur, noise, overexposure, compression artifacts, and other quality issues."
>
> The results are shown below.
>
> | Category  | GPT-5-main | Two-stage reasoning |
> |------------|-------------|---------------------|
> | Emotion    | **92.7** | 87.8 |
> | Existence  | **75.4** | 63.2 |
> | Quality    | 84.5 | **84.6** |
>
> For emotion and existence, two-stage reasoning performed less effectively, consistent with the findings reported in MLLM-CompBench. Interestingly, performance on quality remained nearly identical, suggesting that explicit 1–10 rating prompts can serve as a reliable intermediate representation for comparative judgment in this category.
>
> `Q2. Did annotators label all the data? How many annotators were involved in building this benchmark, and what were their backgrounds?`
>
> We appreciate your comment. In total, 10 annotators were involved in building this benchmark. All annotators are machine learning researchers or engineers with relevant experience in data annotation and model evaluation.
>
> ---
>
> [1] Zhou et al,, VLM4D: Towards Spatiotemporal Awareness in Vision Language Models, ICCV 2025.
>
> [2] Cheng et al., SpatialRGPT: Grounded Spatial Reasoning in Vision‐Language Models, NeurIPS 2024.
>
> [3] Lin et al., CameraBench: Towards Understanding Camera Motions in Any Video, arXiv 2025.

---

### Official Review · Reviewer_Qbbf · 2025-11-02

**Soundness:** 3
**Presentation:** 3
**Contribution:** 4
**Rating:** 6
**Confidence:** 3

**Summary:**

The paper introduces VLM-SubtleBench, a benchmark targeting subtle comparative reasoning across 10 difference types (Attribute, State, Emotion, Temporal, Spatial, Existence, Quantity, Quality, Viewpoint, Action) and 6 domains (Natural, Game, Industry, Aerial, Medical, Synthetic), comprising 11.7k (image-pair, question, answer) triplets. It emphasizes minimally different pairs (e.g., pairs with high DINOv3 similarity) and augments MCQ with a captioning track. Systematic studies span open-source and proprietary VLMs, prompt/fusion strategies, and controlled synthetic stress tests. Human accuracy is ~95.5%, whereas the best proprietary model remains well below this, with particular weaknesses in temporal/spatial/viewpoint categories.

**Strengths:**

- Substantive contribution via task definition & data curation. Clearly formalizing subtle comparative reasoning as an evaluation target and curating a benchmark dataset with transparent collection/validation protocols is, by itself, a meaningful research contribution.

- Breadth + diagnostics. Coverage of 10 difference types and 6 domains with controlled synthetic factors (e.g., brightness deltas, object size, translation, object count) supports failure-mode analysis rather than aggregate scores only.

**Weaknesses:**

- Data generation dependencies. Some Attribute pairs are created with Gemini-2.5 flash image preview (“nano-banana”) editing; Medical questions are refined by gpt-4o. This can introduce stylistic artifacts or distribution shifts that confound evaluation unless carefully audited. Please quantify any such effects (e.g., edited vs. non-edited subsets).

- The paper identifies notable gaps in temporal/spatial/viewpoint (e.g., stable accuracy requiring ~160 px camera translation in synthetic tests), but a more explicit reporting of difficulty curves on natural data would help establish ecological validity.

**Questions:**

- Your Figure 5 synthetic-control study probes failure modes by manipulating low-level factors (e.g., brightness/scale/count/translation). Could you extend this with a color-sensitivity axis inspired by VLM’s Eye Examination [1], which reports consistent green insensitivity across VLMs, to refine the Attribute subset? Concretely, consider hue sweeps with a green vs. non-green contrast and ΔE/brightness steps, and report performance as a function of hue as well as factor interactions (color × size/count/translation). This would test whether known perceptual color deficits compound subtle comparative reasoning failures.

- You clearly motivate the benchmark with high-stakes domains (industrial anomaly detection, medical imaging, aerial surveillance). Could you provide evidence of transfer, e.g., correlations between SubtleBench category scores and downstream metrics on representative datasets in those domains, or a small-scale finetuning study showing measurable gains in real applications?

[1] VLM’s Eye Examination: Instruct and Inspect
Visual Competency of Vision Language Models, https://arxiv.org/pdf/2409.14759

**Details Of Ethics Concerns:**

-

---

> ### Author Response · Authors · 2025-11-21
> **Official Comment by Authors (1/3)**
>
> We truly appreciate the reviewer’s insightful remarks and positive response to our study.
>
> `W1. Data generation dependencies. Some Attribute pairs are created with Gemini-2.5 flash image preview (“nano-banana”) editing; Medical questions are refined by gpt-4o. This can introduce stylistic artifacts or distribution shifts that confound evaluation unless carefully audited. Please quantify any such effects (e.g., edited vs. non-edited subsets).`
>
> We thank the reviewer for raising this important point regarding data generation dependencies. To quantify whether synthetic data generation could bias evaluation, we compared model performance between real and nano-banana-generated counterparts for the same image pairs.
>
> **Setup.** We conducted an experiment using the subset of our benchmark (~10%) for which we had previously annotated difference captions. For each original pair, synthetic counterpart was generated by prompting nano-banana to reconstruct the image, preserving scene content while introducing potential stylistic differences. We evaluated on both the multiple-choice questions and the captioning task (CSS, LLM-Judge).
>
> | Type      | Accuracy | CSS    | LLM-Judge |
> |------------|-----------|--------|---------------|
> | Real | 60.6 | 0.51 | 26.3          |
> | Reconstructed  | 60.6  | 0.51 | 27.3      |
>
> Across all metrics, the differences were negligible, indicating that the use of nano-banana editing does not introduce measurable stylistic or distributional bias that could confound evaluation.
>
>
> `W2. The paper identifies notable gaps in temporal/spatial/viewpoint (e.g., stable accuracy requiring ~160 px camera translation in synthetic tests), but a more explicit reporting of difficulty curves on natural data would help establish ecological validity.`
>
> We thank the reviewer for the constructive suggestion to extend our controlled difficulty analysis from synthetic to real-world settings. Following this feedback, we newly annotated and analyzed real samples for temporal, spatial, and viewpoint differences.
>
> **Setup.** To obtain natural difficulty curves, we constructed three small-scale real datasets:
> - Temporal and Spatial datasets were newly derived from the VLM4D [1] videos. For temporal analysis, we leveraged existing frame-level timestamps to extract new frame pairs with systematically varied temporal intervals. For spatial differences, we manually annotated the object of interest positions in consecutive frames, computing displacement in pixels to define difficulty levels. For both the temporal and spatial cases, we generated the corresponding VQA instances using the image pairs and question–answer annotations provided in each VLM4D video. We divided samples into five magnitude bins: temporal intervals of 0.25s over the range [0s, 1.25s], and spatial displacements of 60 px over the range [0px, 300px], where shorter side of image is 480px. Each intervals have more than 80 pairs
> - For Viewpoint, where accurate camera pose annotation is challenging on real videos, we used 360° HDRI captures of static scenes from Poly Haven [2]. We rendered controlled camera rotations to obtain paired images of identical scenes but with different viewpoints. Specifically, for horizontal (pan) and vertical (tilt), we rotated by [2.5°, 15°] in 2.5° increments, and for roll, by [10°, 60°] in 10° increments, each with 100 pairs per interval. Each question asked: "In which direction does the camera {pan/tilt/roll} from the first image to the second image?" with answer options (left/right), (up/down), or (clockwise/counterclockwise).
>
> **Results.** We present the evaluation result in the table below. Across all three dimensions, the model exhibits interpretable difficulty-dependent behavior consistent with our synthetic analysis.
>
> | Magnitude bin    | 1   | 2   | 3   | 4   | 5   |
> |-----|-----|-----|-----|-----|-----|
> | Temporal | 54.3 | 56.5 | 60.5 | 64.4 | 65.8 |
> | Spatial | 59.0 | 64.5 | 66.5 | 66.9 | 66.7 |
> | Viewpoint (Pan) | 65.0 | 75.0 | 83.0 | 96.0 | 92.0 |
> | Viewpoint (Tilt) | 78.0 | 90.0 | 92.0 | 99.0 | 98.0 |
> | Viewpoint (Roll) | 51.0 | 46.0 | 51.0 | 44.0 | 58.0 |
>
> Accuracy for temporal variations increases steadily from 54.3% to 65.8% as interval length grows from 0s to 1.25s, with the largest gains observed at shorter intervals where small increases in temporal gap significantly enhance the use of motion continuity cues. Beyond roughly 1s, performance saturates as larger scene and context changes begin to reduce temporal coherence. For spatial displacement, accuracy rises from 59.0% to 66.9% and then remains stable, indicating that moderate spatial shifts are sufficient for the model to exploit positional cues, while larger displacements no longer yield additional benefits. For viewpoint differences, both pan and tilt show clear monotonic gains, reaching near-perfect accuracy (~95–99%) at 10–12.5° rotations. Howver, roll remains close to random even at 50–60°, suggesting limited robustness to in-plane rotation.

---

> ### Author Response · Authors · 2025-11-21
> **Official Comment by Authors (2/3)**
>
> `Q1. Your Figure 5 synthetic-control study probes failure modes by manipulating low-level factors (e.g., brightness/scale/count/translation). Could you extend this with a color-sensitivity axis inspired by VLM’s Eye Examination [1], which reports consistent green insensitivity across VLMs, to refine the Attribute subset? Concretely, consider hue sweeps with a green vs. non-green contrast and ΔE/brightness steps, and report performance as a function of hue as well as factor interactions (color × size/count/translation). This would test whether known perceptual color deficits compound subtle comparative reasoning failures.`
>
> We thank the reviewer for this excellent suggestion. Following your comment, we extended our synthetic-control analysis with a color-sensitivity axis to examine whether hue-level perceptual biases compound subtle comparative reasoning failures.
>
> **Setup.** We prefixed five representative colors (two green tones and three non-green: blue, red, and magenta) and systematically varied ΔE (hue shift in OKLAB space), brightness (*L* channel), size, count, and translation. We used OKLAB, which aligns well with human perceptual uniformity: *L* encodes brightness, while *a* and *b* correspond to green–magenta and blue–yellow opponent axes. To isolate hue effects, ΔE adjustments were applied by modifying hue (*a*, *b*) while keeping brightness *L* constant. Brightness changes fixed hue but sampled L values within [0.4, 0.8]. All other setup conditions followed Figure 5.
>
> **Results.** For an intuitive display of the results, **the results are presented as heatmaps in Appendix C.4**.
> Consistent with prior work (VLM’s Eye Examination), models showed marked difficulty distinguishing green hues, with significantly lower accuracy compared to red or blue. Magenta exhibited the strongest degradation (near 0%), revealing a systematic color-specific weakness in GPT-4o. Brightness variation produced no notable color-dependent gap, suggesting that VLMs are relatively color-invariant when judging luminance. Cross-factor analyses (color vs. size/count/viewpoint) showed minimal interaction, implying that these tasks are less influenced by color sensitivity because hue discrimination itself plays a minor role.

---

> ### Author Response · Authors · 2025-11-21
> **Official Comment by Authors (3/3)**
>
> `Q2. You clearly motivate the benchmark with high-stakes domains (industrial anomaly detection, medical imaging, aerial surveillance). Could you provide evidence of transfer, e.g., correlations between SubtleBench category scores and downstream metrics on representative datasets in those domains, or a small-scale finetuning study showing measurable gains in real applications?`
>
> **Setup.** To demonstrate that our benchmark correlates with real-world applications, we measured the relationship between SubtleBench performance and downstream metrics on an established real-world anomaly detection benchmark, MMAD [3]. MMAD is specifically designed to evaluate multimodal large language models (MLLMs) on industrial anomaly detection tasks. We focused on the anomaly discrimination task of MMAD, which assesses a model’s ability to distinguish anomalous from normal samples. We used a randomly sampled 10% subset of MMAD. Using the same set of ten open-source and proprietary models reported in Table 2, we computed Spearman correlations [4] between SubtleBench scores and MMAD discrimination accuracy. As a baseline, we also computed the correlation between MMAD and another benchmark, MLLM-CompBench.
>
> | Models              | Ours | MLLM-CompBench | MMAD |
> |----------------------|------|----------------|------|
> | Qwen-2.5-VL-7B       | 59.4 | 73.6           | 65.0 |
> | Qwen-2.5-VL-32B      | 62.2 | 74.6           | 67.6 |
> | Qwen-2.5-VL-72B      | 65.4 | 76.9           | 68.9 |
> | GPT-4o               | 61.6 | 75.7           | 67.7 |
> | o3                   | 75.7 | 86.3           | 72.9 |
> | GPT-5-main           | 71.3 | 83.9           | 70.6 |
> | GPT-5-thinking       | 77.8 | 86.3           | 73.5 |
> | Claude-sonnet-4      | 62.6 | 73.6           | 70.9 |
> | gemini-2.5-flash     | 62.5 | 85.2           | 71.4 |
> | gemini-2.5-pro       | 68.2 | 87.2           | 72.2 |
>
> - corr(VLM-SubtleBench, MMAD) = 0.8424
> - corr(MLLM-CompBench, MMAD) = 0.8110
>
> **Results.** The analysis reveals a strong positive correlation between our benchmark and MMAD, supporting the transferability of subtle comparative reasoning to real-world anomaly detection tasks. Specifically, the correlation between SubtleBench and MMAD is ρ = 0.8424 (p = 0.0022), while MLLM-CompBench and MMAD showed ρ = 0.8110 (p = 0.0044). These results indicate that models performing well on SubtleBench tend to exhibit superior performance on applied anomaly detection benchmarks, reinforcing that our benchmark captures reasoning capabilities that are predictive of success in high-stakes, real-world scenarios.
>
> We further plan to extend this analysis during the rebuttal period by examining correlations with benchmarks from other high-stakes domains (e.g., game and aerial surveillance) to validate the generality of the observed transfer patterns.
>
> ---
>
> [1] Zhou et al,, VLM4D: Towards Spatiotemporal Awareness in Vision Language Models, ICCV 2025.
>
> [2] Poly Haven, https://polyhaven.com/hdris.
>
> [3] Jiang et al., MMAD: A Comprehensive Benchmark for Multimodal Large Language Models in Industrial Anomaly Detection, ICLR 2025.
>
> [4] C. Spearman, The proof and measurement of association between two things, The American Journal of Psychology, 1904.

---

> ### Author Response · Authors · 2025-12-03
> **Additional Response Regarding Q2**
>
> We thank again the reviewer for motivating further evidence of transfer to high-stakes domains. Following the feedback, we conducted (1) additional fine-tuning experiments on industrial anomaly detection (MMAD) and (2) extended correlation and fine-tuning analyses to the aerial surveillance domain.
>
> **Correlation Study.** To assess whether our benchmark meaningfully reflects downstream perceptual reasoning in high-stakes domains, we extended our correlation analysis to the aerial surveillance setting. We used QAG-360K [1], a large-scale aerial change-detection VQA dataset providing paired remote-sensing images with eight question categories covering object- and region-level changes. We excluded the change ratio category because its answers are continuous percentage ranges, making multiple-choice question (MCQ) conversion ambiguous. We randomly sampled 723 validation examples and computed Spearman correlations between model performance on QAG-360K and performance on both VLM-SubtleBench and MLLM-CompBench.
>
> The resulting cross-benchmark alignment is strong:
> - corr(VLM-SubtleBench, QAG-360K) = 0.7212
> - corr(MLLM-CompBench, QAG-360K) = 0.7195
>
> These results demonstrate that the fine-grained perceptual differences measured by our benchmark reliably reflect capabilities relevant to real-world aerial change detection.
>
> **Fine-tuning Study.** To further examine transferability, we conduct fine-tuning experiments in the applied domains.  We fine-tuned Qwen-2.5-VL-7B on newly-curated training split of VLM-SubtleBench (1,277 samples) and, for comparison, on a matched-size subset of MLLM-CompBench test set. For MMAD, all training instances were standardized to the MMAD task format; for QAG-360K, both training and evaluation were standardized to our base prompt as QAG-360K does not offer a prompt format. Aside from this domain-specific formatting, the fine-tuning procedure was identical across datasets.
>
>
> | Model (Qwen-2.5-VL-7B)       | MMAD Accuracy | QAG-360K Accuracy |
> | ---------------------------- | ------------- | ----------------- |
> | Base | 65.0 | 34.4 |
> | Fine-tuned on Ours | **69.6** | **35.5** |
> | Fine-tuned on MLLM-CompBench | 66.3 | 32.2 |
>
> In industrial anomaly detection, fine-tuning on our dataset yields the largest gains over the base model (+4.6), indicating that the subtle perceptual cues emphasized by SubtleBench directly benefit real-world anomaly reasoning. In aerial surveillance, fine-tuning on our dataset improves accuracy, while fine-tuning on MLLM-CompBench decreases it, demonstrating that SubtleBench’s comparative signals are more domain-relevant for remote-sensing change detection.
>
> [1] li et al., Show Me What and Where has Changed? Question Answering and Grounding for Remote Sensing Change Detection, 2024.

---

### Official Review · Reviewer_chNu · 2025-11-03

**Soundness:** 3
**Presentation:** 3
**Contribution:** 3
**Rating:** 6
**Confidence:** 3

**Summary:**

The paper proposes a benchmark to evaluate how well models are at discerning subtle changes between two images in form of visual question answering and captioning. The dataset covers instances from multiple domains (such as game, industrial, medical, etc) and has ten different types of differences (such as temporal, spatial, emotion, etc). Evaluation results are comprehensive including multiple popular open-source and proprietary models such as GPT-5, o3, Claude-sonnet-4, Gemini-2.4-pro along with different prompting strategies. Human evaluation is also conducted and current results indicate a gap between human performance (avg. performance 95.5) and current frontier models (best 77.8 avg performance).

**Strengths:**

1. The paper focuses on an interesting problem setup of fine-grained changes between two images, and it is interesting how current frontier models struggle at these tasks.
2. The paper adequately describes the dataset construction process, model evaluation setup, and experimental results. Overall, it is well written.
3. The evaluation process studies multiple factors that can influence model performance -- such as how to combine the two images when feeding images, different prompting strategies and impact on model performance with diff. controllable percentage of change b/w 2 images.

**Weaknesses:**

1. This task of subtle difference changes b/w two images has been previously explored in works such as Spot-the-Diff [1], Img-Diff [2] and MLLM-CompBench [3] as noted by authors. The primary novelty seems to be expansion to multiple domains, more question types and combination of multiple choice questions and captioning in a single benchmark. In this regard, novelty is a bit limited.

2. There can be further baselines/prompting strategies considered such as:
- Calculating regions of interest from the subtraction of 2 images, and then highlighting these regions in the 2 input images (through simple bounding boxes or masks) and feed them to VLM stating regions of interest are highlighted.
- 2-step reasoning process -- first ask the VLM to describe differences b/w the 2 images with respect to answering the question, and then feed this output in addition to the 2 images and original question.

Relatively minor:
3. It would also be interesting to study whether this task can simply be solved by training models on a mix of synthetic and real samples as the task itself might be out-of-distribution. But I understand this may be out of scope for current paper.

**Questions:**

Please see weaknesses.

---

> ### Author Response · Authors · 2025-11-21
> **Official Comment by Authors (1/2)**
>
> We are deeply grateful to the reviewer for the thoughtful comments and encouraging evaluation of our manuscript.
>
> `W1. This task of subtle difference changes b/w two images has been previously explored in works such as Spot-the-Diff [1], Img-Diff [2] and MLLM-CompBench [3] as noted by authors. The primary novelty seems to be expansion to multiple domains, more question types and combination of multiple choice questions and captioning in a single benchmark. In this regard, novelty is a bit limited.`
>
> We appreciate the reviewer’s thoughtful observation. VLM-SubtleBench introduces a systematic evaluation of fine-grained, visually and semantically subtle differences across multiple domains and difference types, enabling more interpretable and diagnostic assessments of VLM reasoning beyond what prior comparative benchmarks capture.
>
> | Models              | Ours | MLLM-CompBench | MMAD |
> |----------------------|------|----------------|------|
> | Qwen-2.5-VL-7B       | 59.4 | 73.6           | 65.0 |
> | Qwen-2.5-VL-32B      | 62.2 | 74.6           | 67.6 |
> | Qwen-2.5-VL-72B      | 65.4 | 76.9           | 68.9 |
> | GPT-4o               | 61.6 | 75.7           | 67.7 |
> | o3                   | 75.7 | 86.3           | 72.9 |
> | GPT-5-main           | 71.3 | 83.9           | 70.6 |
> | GPT-5-thinking       | 77.8 | 86.3           | 73.5 |
> | Claude-sonnet-4      | 62.6 | 73.6           | 70.9 |
> | gemini-2.5-flash     | 62.5 | 85.2           | 71.4 |
> | gemini-2.5-pro       | 68.2 | 87.2           | 72.2 |
>
> - corr(VLM-SubtleBench, MMAD) = 0.8424
> - corr(MLLM-CompBench, MMAD) = 0.8110
>
> Importantly, the covered domains—including industrial inspection, aerial surveillance, medical imaging, and gaming environments—represent practical scenarios where subtle comparative reasoning is critical. To further substantiate this real-world relevance, we conducted a **correlation analysis between our benchmark and MMAD [2]**, an established anomaly detection benchmark. The ranking **correlation between SubtleBench and MMAD** (0.8424, p = 0.0022) **exceeds that between MLLM-CompBench and MMAD** (0.8110, p = 0.0044), indicating that improvements on SubtleBench more reliably translate into downstream task gains. This alignment with practical performance underscores the novelty and value of our benchmark that bridges controlled evaluation and real-world VLM reasoning.
>
> Note that, we aim to use the remaining rebuttal period to broaden our correlation analysis and better connect our benchmark to real-world use cases.
>
> ---
> [1] Jiang et al., MMAD: A Comprehensive Benchmark for Multimodal Large Language Models in Industrial Anomaly Detection, ICLR 2025.
>
> [2] C. Spearman, The proof and measurement of association between two things, The American Journal of Psychology, 1904.

---

> ### Author Response · Authors · 2025-11-21
> **Official Comment by Authors (2/2)**
>
> `W2. There can be further baselines/prompting strategies considered such as: (1) Calculating regions of interest from the subtraction of 2 images, and then highlighting these regions in the 2 input images (through simple bounding boxes or masks) and feed them to VLM stating regions of interest are highlighted. (2) 2-step reasoning process -- first ask the VLM to describe differences b/w the 2 images with respect to answering the question, and then feed this output in addition to the 2 images and original question.`
>
> We appreciate the reviewer’s insightful suggestions. We implemented both of the proposed strategies:
>
> (1) Highlighting Regions of Interest: We retained pixels with large differences between the two images and grouped adjacent pixels together. Bounding boxes were then drawn around the three largest clusters to highlight the main regions of change. Both the original image pair and the highlighted versions were included in the prompt to the VLM.
>
> (2) Two-Step Reasoning: In the first stage, we asked the VLM to provide a detailed description of the differences between the two images while informing it that the next question would be based on this description. The resulting description and answer from the first stage were then inserted into the prompt for the second stage.
>
> | Category | GPT-5-main | highlight | 2-step reasoning |
> |------------|-------------|----------------|-----------------------|
> | Attribute | 75.4 | 71.1 | 70.8 | | State | 78.4 | 75.2 | 79.1 |
> | Emotion | 92.7 | 92.0 | 93.4 | | Temporal | 53.6 | 51.1 | 56.9 |
> | Spatial | 50.1 | 54.9 | 47.4 | | Existence | 75.4 | 86.5 | 81.3 |
> | Quantity | 72.6 | 74.3 | 66.4 | | Quality | 84.5 | 77.8 | 83.5 |
> | Camera | 57.5 | 57.3 | 58.0 | | Action | 83.6 | 82.9 | 83.6 |
> | **Overall** | 71.3 | **71.5** | 71.0 |
>
> For the highlighting approach, performance improved slightly overall. It performed well on datasets with limited changes (e.g., synthetic data), but struggled on datasets with strong variations in brightness or image quality (e.g., YT8M), where bounding boxes often failed to correctly capture the change regions.
>
> For the two-step reasoning approach, performance decreased slightly. In many cases, the model produced descriptions claiming “no difference” in the first stage, leading to incorrect final answers. Unlike the single-step reasoning setting—where reasoning and answering occur together- the performance dropped
>
> We will include these additional results and analysis in the Prompting Strategies section of the revised manuscript.
>
> `W3. It would also be interesting to study whether this task can simply be solved by training models on a mix of synthetic and real samples as the task itself might be out-of-distribution. But I understand this may be out of scope for current paper.`
>
> We thank reviewer for valuable feedback. To examine whether data alone can mitigate the comparative reasoning challenge, we curated an additional training dataset following the same collection pipeline as our test set. Specifically, we constructed a dataset amounting to 10% of the original test-set size (~1k samples).
>
> When fine-tuning Qwen-2.5-VL-7B on this dataset, we observed consistent improvements across all difference types, as summarized below:
>
> | Difference Type | Qwen-2.5-VL-7B | Qwen-2.5-VL-7B (fine-tuned) | GPT-5-thinking | Human Eval
> |-----|-----|----|----|----|
> | Attribute   | 46.5           | 62.0  | 83.6 | 92.0
> | State           | 63.7           | 69.1    | 80.7 | 93.0
> | Emotion         | 87.8           | 92.2   | 93.1|93.0
> | Temporal        | 50.2           | 52.5   | 60.2|93.0
> | Spatial         | 39.5           | 47.0    | 59.9|95.0
> | Existence       | 73.8           | 85.3    |85.4|97.0
> | Quantity        | 58.0           | 77.0  |79.9|97.0
> | Quality         | 70.9           | 85.9   |84.8|99.0
> | Camera          | 47.5           | 57.5  |68.5|98.0
> | Action          | 69.3           | 75.4    |84.9|98.0
>
> Fine-tuning on in-distribution data lead to moderate gain across most categories, particularly in existence, quantity, and quality. However, improvements were relatively smaller for spatial and temporal categories, which likely require richer spatial–temporal reasoning rather than distributional adaptation. Even after fine-tuning, there remains a significant performance gap compared to GPT-5-thinking and human. Leveraging additional data to further enhance generalization and surpass proprietary models presents an exciting direction for future research beyond the scope of this study.
>
> **Note**. We identified an issue in the originally reported results for Qwen-2.5-VL-7B and have corrected them accordingly. During evaluation via an external API service (OpenRouter), we found that the 7B model frequently returned empty responses without any error messages, which affected the initially submitted results. All numbers reported here reflect the corrected evaluation.

---

> > ### Comment · Reviewer_chNu · 2025-11-27
> >
> > Thank you for the clarifications and updated results including finetuning results which can be useful to add to main paper/appendix.

---

> > > ### Author Response · Authors · 2025-11-27
> > >
> > > We are glad that our rebuttal has clarified your concerns, including the fine-tuning results. If you have any further questions, please feel free to let us know. We are more than happy to address them during the remaining rebuttal period. Thank you for your time and effort.

---

> ### Author Response · Authors · 2025-12-03
> **Additional Response Regarding W1**
>
> We thank the reviewer again for raising this important point regarding the novelty of our benchmark. To further substantiate the real-world relevance of our benchmark, we provide additional evidence showing that the fine-grained comparative skills measured by VLM-SubtleBench reliably transfer to high-stakes domains and thereby demonstrate contributions beyond prior comparative datasets.
>
> **Correlation Study.** To evaluate whether VLM-SubtleBench meaningfully reflects downstream perceptual reasoning demands, we further examined cross-benchmark alignment with a real-world **aerial surveillance** task. Using QAG-360K [1], a large-scale change-detection benchmark, we evaluated the same ten models used in our MMAD correlation analysis and measured Spearman correlations with both SubtleBench and MLLM-CompBench. The results show substantial alignment:
> - corr(VLM-SubtleBench, QAG-360K) = 0.7212
> - corr(MLLM-CompBench, QAG-360K) = 0.7195
>
> While both benchmarks correlate with QAG-360K, SubtleBench exhibits stronger alignment, suggesting that the fine-grained comparative skills it measures are more predictive of performance on real-world, high-stakes perceptual tasks.
>
> **Fine-tuning Study.** To further assess transferability, we also conducted fine-tuning experiments in two applied domains: industrial anomaly detection and aerial surveillance. We fine-tuned Qwen-2.5-VL-7B on newly-curated training split of VLM-SubtleBench (1,277 samples) and, for comparison, on a matched-size subset of MLLM-CompBench test set. For MMAD, all training instances were standardized to the MMAD task format; for QAG-360K, both training and evaluation were standardized to our base prompt as QAG-360K does not offer a prompt format. Aside from this domain-specific formatting, the fine-tuning procedure was identical across datasets.
>
> | Model (Qwen-2.5-VL-7B)       | MMAD     | QAG-360K |
> | ---------------------------- | -------- | -------- |
> | Base                         | 65.0     | 34.4     |
> | Fine-tuned on Ours           | **69.6** | **35.5** |
> | Fine-tuned on MLLM-CompBench | 66.3     | 32.2     |
>
> In industrial anomaly detection, fine-tuning on our dataset yields the largest gains over the base model (+4.6), indicating that the subtle perceptual cues emphasized by SubtleBench directly benefit real-world anomaly reasoning. In aerial surveillance, fine-tuning on our dataset improves accuracy, while fine-tuning on MLLM-CompBench decreases it, demonstrating that SubtleBench’s comparative signals are more domain-relevant for remote-sensing change detection.
>
> **Implication for novelty.** These findings reinforce that our benchmark is not merely a broader extension of prior work. Rather, its design choices, including the use of **multi-domain and fine-grained difference types**, enable the evaluation of perceptual reasoning skills that **transfer effectively to real-world, high-stakes tasks**. The ability of VLM-SubtleBench to capture fine-grained reasoning skills that are reflected in downstream performance provides concrete evidence of its practical and scientific value beyond existing comparative benchmarks.
>
> [1] Li et al., Show Me What and Where has Changed? Question Answering and Grounding for Remote Sensing Change Detection, 2024.

---

### Official Review · Reviewer_Waq7 · 2025-11-03

**Soundness:** 2
**Presentation:** 3
**Contribution:** 2
**Rating:** 2
**Confidence:** 3

**Summary:**

In this paper, the author presented VLM-SubtleBench, a benchmark to evaluate subtle comparative reasoning in VLMs across 10 difference types and 6 visual domains. The benchmark contains more than 11k image-pair QA samples and 1k human-annotated difference captions. In this paper, the authors also claim that existing such benchmarks focus on salient differences and lack domain diversity. They evaluated several current VLMs on the benchmark and found that they struggle with subtle differences compared to human performance.

**Strengths:**

- In this paper, they introduced an increasingly relevant capability: subtle visual comparison between images, across multiple domains. It contains various difference types (attribute, temporal, viewpoint, etc.) and datasets beyond natural images (industrial, medical, etc.).
- It also has a mix of real data and synthetic setups, to show controlled evaluation capabilities.
- The paper is easy to read, though the figures could use more text to make them clearer.

**Weaknesses:**

- The dataset seems to be largely a small increment of prior multi-image VLM benchmarks like MLLM-CompBench, ReMI. The claim of novelty in subtlety is only partially convincing.  Subtle differences are defined via embedding cosine similarity (DINOv3), but this does not necessarily guarantee perceptual or semantic subtlety.
- In Figure 3, it can be seen that when the catcher moved, the other player moved as well. As the paper claims to be fine-grained, I would be interested to know how the authors ensured that in the video, only the object/person in question moved, not the other parts.
- Repetition of the same datasets across categories. MVTEC-AD reused across “attribute” and “state” categories.
- “Overlap” and “subtraction” images are described vaguely. It is unclear to me whether pixel-level overlay vs feature-space subtraction is used. This step needs technical clarity and justification. [line 354]
- No evaluation on two-image native models (e.g., LLaVA Next, Clip).
- What is domain domain-specific performance difference for each model in the benchmark? Can the author provide some insights on: are the methods bad at medical or really good in synthetic for every type?
- Typos: line 242, archange to are

**Questions:**

- how subtlety was maintained for the types from frames were takes from videos, to be sure that only the object in question moved.
- How do authors ensure repetition of the same datasets across categories represents distinct types [weakness 3]?
- Insights about domain-specific evaluation and evaluating multi-image models on the benchmark [weakness 5].

---

> ### Author Response · Authors · 2025-11-21
> **Official Comment by Authors (1/3)**
>
> We appreciate the reviewer’s constructive feedback and helpful suggestions.
>
> `W1. The dataset seems to be largely a small increment of prior multi-image VLM benchmarks like MLLM-CompBench, ReMI. The claim of novelty in subtlety is only partially convincing. Subtle differences are defined via embedding cosine similarity (DINOv3), but this does not necessarily guarantee perceptual or semantic subtlety.`
>
> **Novelty.** We emphasize that subtle comparative reasoning is a **highly important yet underexplored** capability for VLMs. In many real-world applications involving advanced comparative cognition skills, **achieving a certain level of subtlety is crucial**. For example, in anomaly detection or aerial surveillance, success hinges on detecting differences below a fine pixel/unit threshold. Without such sensitivity, VLMs' performance collapses toward randomness (as shown in most cases of Table 2). Yet, prior benchmarks (e.g., MLLM-CompBench and ReMI) either ignore this subtlety  or cover only narrow domains (i.e., natural images), leaving a clear gap from real applications.
>
> | Models              | Ours | MLLM-CompBench | MMAD |
> |----------------------|------|----------------|------|
> | Qwen-2.5-VL-7B       | 59.4 | 73.6           | 65.0 |
> | Qwen-2.5-VL-32B      | 62.2 | 74.6           | 67.6 |
> | Qwen-2.5-VL-72B      | 65.4 | 76.9           | 68.9 |
> | GPT-4o               | 61.6 | 75.7           | 67.7 |
> | o3                   | 75.7 | 86.3           | 72.9 |
> | GPT-5-main           | 71.3 | 83.9           | 70.6 |
> | GPT-5-thinking       | 77.8 | 86.3           | 73.5 |
> | Claude-sonnet-4      | 62.6 | 73.6           | 70.9 |
> | gemini-2.5-flash     | 62.5 | 85.2           | 71.4 |
> | gemini-2.5-pro       | 68.2 | 87.2           | 72.2 |
>
> - corr(VLM-SubtleBench, MMAD) = 0.8424
> - corr(MLLM-CompBench, MMAD) = 0.8110
>
> Furthermore, to quantify our benchmark's real-world relevance, we **additionally conducted a correlation analysis with downstream benchmarks**. Specifically, we calculate the ranking correlation [1] of 10 VLMs' performance between MMAD [2] (an anomaly detection benchmark) and our SubtleBench, and compare it with that between MMAD and MLLM-CompBench. As shown in the table above, the ranking **correlation between SubtleBench and MMAD** (0.8424 with p = 0.0022) **is significantly higher than that between MLLM-CompBench and MMAD** (0.8110 with p = 0.0044). This implies that improving performance on our SubtleBench is more likely to translate into real-task gains, which demonstrate clear practical value and novelty.
>
> Note that, we plan to extend this correlation study during the remaining rebuttal period to further show how our benchmark narrows the gap toward real-world applications.
>
> **Definition of Subtlety.** We apologize for the confusion. Subtlety is *not* defined via cosine similarity (DINOv3). Our notion of "subtle difference" **covers both semantic and perceptual nuances**, rather than being limited to numerical similarity. Concretely, we curate image pairs that exhibit visually or semantically minor yet meaningful differences across ten categories, ranging from low-level variations (e.g., color, texture, or spatial shifts) to high-level concepts (emotion, action, or viewpoint). The DINOv3 cosine similarity was not intended to define subtlety but to quantify and control it during dataset construction, serving as a *necessary filtering condition* to ensure visual similarity between pairs. We will clarify this in the revised manuscript.
>
> ---
> [1] C. Spearman, The proof and measurement of association between two things, The American Journal of Psychology, 1904.
>
> [2] Jiang et al., MMAD: A Comprehensive Benchmark for Multimodal Large Language Models in Industrial Anomaly Detection, ICLR 2025.

---

> ### Author Response · Authors · 2025-11-21
> **Official Comment by Authors (2/3)**
>
> `W2&Q1. In Figure 3, it can be seen that when the catcher moved, the other player moved as well. As the paper claims to be fine-grained, I would be interested to know how the authors ensured that in the video, only the object/person in question moved, not the other parts.`
>
> We appreciate the reviewer’s insightful observation. We **intentionally did not ensure** other parts in the scene to remain static when generating image pairs. This was a deliberate design choice, and we view it as a strength rather than a limitation for three reasons:
>
> 1. **Closer to real-world practice**.
> In real applications (e.g., surveillance, industrial inspection), changes in the target of interest often co-occur with background or contextual changes. Ensuring that only one entity changes while all others remain fixed is rarely feasible in practice. Thus, including such incidental variations makes the benchmark more representative and ecologically valid.
> 2. **Higher task difficulty**.
> Allowing other parts to move introduces visual distractions that make the comparison more challenging, requiring models to focus selectively on the semantically relevant changes. This prevents subtle comparison tasks from being easily exploited by simple tricks such as overlay or subtraction.
> 3. **Benchmark robustness**.
> Because multiple aspects may change simultaneously, the benchmark can evaluate whether models truly understand which difference is being asked about. To further diversify the expressions, our benchmark supports both multiple-choice and captioning questions.
>
> In summary, we intentionally did not enforce perfect isolation of motion or change, as doing so would produce an overly sanitized setting that diverges from real-world conditions. Instead, we prioritize practical realism and robustness, making the benchmark both more challenging and more meaningful.
>
> `W3&Q2. Repetition of the same datasets across categories. MVTEC-AD reused across “attribute” and “state” categories.`
>
> We thank the reviewer for the feedback. To clarify, while MVTEC-AD is used in both the attribute and state categories, **no image pair, defect type, or question overlaps between them**. MVTEC-AD provides defect-level annotations across diverse anomaly types, which we leveraged to separate state-related defects (e.g., damaged transistor cases, bent nuts) from attribute-related ones (e.g., discolored leather, rough-surfaced zippers). Thus, each category reflects a different dimension of subtle visual reasoning.
>
> `W4. “Overlap” and “subtraction” images are described vaguely. It is unclear to me whether pixel-level overlay vs feature-space subtraction is used. This step needs technical clarity and justification. [line 354]`
>
> We thank the reviewer for highlighting this point. Both the overlap and subtraction are conducted in the pixel space. Specifically, the *overlap* image is created by blending the two spatially aligned inputs with equal pixel-wise weights (50% each), producing a composite image that visually combines the two. The *subtraction* image is obtained by taking the absolute pixel-wise difference between two inputs, converting the result to grayscale, and normalizing it to emphasize regions of maximal change. Both images are provided to the VLM along with the originals, enabling explicit visual grounding of subtle differences. The manuscript has been updated accordingly (see lines 353–358).

---

> ### Author Response · Authors · 2025-11-21
> **Official Comment by Authors (3/3)**
>
> `W5&Q3. No evaluation on two-image native models (e.g., LLaVA Next, Clip).`
>
> We appreciate the reviewer’s suggestion to include models that are natively designed for multi-image comparison. We have **additionally evaluated LLaVA-Next and LLaVA-OneVision, and present their results** below alongside Qwen2.5-VL-7B. The experiments were conducted under the same settings as Table 2 in our paper, and the corresponding results have now been added to Table 2 in the revised manuscript.
>
> | Category   | Qwen2.5-VL-7B | LLAVA-OneVision-7B | LLAVA-NEXT-7B |
> |------------|---------------|--------------------|---------------|
> | Attribute  | **46.5**      | 41.6               | 37.0          |
> | State      | **63.7**      | 56.8               | 51.3          |
> | Emotion    | **87.8**      | 73.9               | 51.8          |
> | Temporal   | **50.2**      | 48.7               | 47.4          |
> | Spatial    | **39.5**      | 35.5               | 37.3          |
> | Existence  | **73.8**      | 44.2               | 25.6          |
> | Quantity   | **58.0**      | 54.9               | 49.5          |
> | Quality    | **70.9**      | 62.7               | 48.0          |
> | Camera     | 47.5          | **49.1**           | 43.7          |
> | Action     | **69.3**      | 60.5               | 46.9          |
>
> Across most categories, Qwen2.5-VL-7B achieved the highest accuracy, followed by LLaVA-OneVision-7B, with LLaVA-Next-7B showing relatively lower performance.
>
> Regarding CLIP, we would like to clarify that its architecture is not inherently designed for question–answering tasks involving two images. A straightforward adaptation would involve reformulating each question–answer pair into descriptive prompts compatible with CLIP and then computing cosine similarities between these text embeddings and the two image embeddings. However, this introduces substantial paraphrasing, making a fair comparison difficult.
>
> `W6. What is domain domain-specific performance difference for each model in the benchmark? Can the author provide some insights on: are the methods bad at medical or really good in synthetic for every type?`
>
> We thank the reviewer for this valuable suggestion. The table below summarizes domain-specific accuracy for proprietary VLMs.
> | Domain  | Random Guess | GPT-4o | o3 | GPT-5-main | GPT-5-thinking | Claude-sonnet-4 | gemini-2.5-flash | gemini-2.5-pro | Average |
> |------------|--------|--------|----|-------------|-------|-------|------------------|----------------|---|
> | natural    | 49.2         | 68.4   | **77.3** | 74.1        | 77.2           | 64.2             | 68.1             | 73.2   | 71.8        |
> | game       | 50.0         | 65.8   | **76.4** | 72.1        | 75.5           | 60.3             | 66.5             | 71.8 | 69.8          |
> | industry   | 50.0         | 71.5   | 79.3 | 77.9        | **81.2**       | 66.3             | 73.7             | 79.7           | 75.7
> | aerial     | 26.9         | 46.9   | 71.4 | 60.8        | 70.7           | 74.1             | 73.4             | **75.7**       | 67.6
> | synthetic  | 29.3         | 45.0   | 72.5 | 63.6        | **78.8**       | 56.3             | 43.3             | 50.6           | 58.6
> | medical    | 50.0         | 62.4   | 68.5 | 78.8        | **82.4**       | 54.8             | 62.4             | 68.8           | 68.3 |
>
> Across models, o3 and GPT-5-thinking achieved the highest scores in most categories. In particular, they exhibited a substantial performance gap over other models in the synthetic and medical domains, while both showed comparatively weaker results on the aerial domain.
>
> Across domains, models generally performed strongest on natural and industry imagery—domains while performance degraded in synthetic and aerial settings.
>
> `W7. Typos: line 242, archange to are`
>
> Thank you for pointing this out. We will correct it in the revised manuscript.

---

> ### Author Response · Authors · 2025-12-03
> **Additional Response Regarding W1**
>
> We thank the reviewer again for raising this important point regarding the novelty of our benchmark. To further substantiate the real-world relevance of our benchmark, we provide additional evidence showing that the fine-grained comparative skills measured by VLM-SubtleBench reliably transfer to high-stakes domains and thereby demonstrate contributions beyond prior comparative datasets.
>
> **Correlation Study.** To evaluate whether VLM-SubtleBench meaningfully reflects downstream perceptual reasoning demands, we further examined cross-benchmark alignment with a real-world **aerial surveillance** task. Using QAG-360K [1], a large-scale change-detection benchmark, we evaluated the same ten models used in our MMAD correlation analysis and measured Spearman correlations with both SubtleBench and MLLM-CompBench. The results show substantial alignment:
> - corr(VLM-SubtleBench, QAG-360K) = 0.7212
> - corr(MLLM-CompBench, QAG-360K) = 0.7195
>
> While both benchmarks correlate with QAG-360K, SubtleBench exhibits stronger alignment, suggesting that the fine-grained comparative skills it measures are more predictive of performance on real-world, high-stakes perceptual tasks.
>
> **Fine-tuning Study.** To further assess transferability, we also conducted fine-tuning experiments in two applied domains: industrial anomaly detection and aerial surveillance. We fine-tuned Qwen-2.5-VL-7B on newly-curated training split of VLM-SubtleBench (1,277 samples) and, for comparison, on a matched-size subset of MLLM-CompBench test set. For MMAD, all training instances were standardized to the MMAD task format; for QAG-360K, both training and evaluation were standardized to our base prompt as QAG-360K does not offer a prompt format. Aside from this domain-specific formatting, the fine-tuning procedure was identical across datasets.
>
> | Model (Qwen-2.5-VL-7B)       | MMAD     | QAG-360K |
> | ---------------------------- | -------- | -------- |
> | Base                         | 65.0     | 34.4     |
> | Fine-tuned on Ours           | **69.6** | **35.5** |
> | Fine-tuned on MLLM-CompBench | 66.3     | 32.2     |
>
> In industrial anomaly detection, fine-tuning on our dataset yields the largest gains over the base model (+4.6), indicating that the subtle perceptual cues emphasized by SubtleBench directly benefit real-world anomaly reasoning. In aerial surveillance, fine-tuning on our dataset improves accuracy, while fine-tuning on MLLM-CompBench decreases it, demonstrating that SubtleBench’s comparative signals are more domain-relevant for remote-sensing change detection.
>
> **Implication for novelty.** These findings reinforce that our benchmark is not merely a broader extension of prior work. Rather, its design choices, including the use of **multi-domain and fine-grained difference types**, enable the evaluation of perceptual reasoning skills that **transfer effectively to real-world, high-stakes tasks**. The ability of VLM-SubtleBench to capture fine-grained reasoning skills that are reflected in downstream performance provides concrete evidence of its practical and scientific value beyond existing comparative benchmarks.
>
> [1] Li et al., Show Me What and Where has Changed? Question Answering and Grounding for Remote Sensing Change Detection, 2024.

---

### Author Response · Authors · 2025-12-03
**General Response**

We sincerely appreciate the reviewers' and ACs' time and effort in serving the ICLR community. Below, we summarize the main review points and our corresponding responses.

Most reviewers agreed that our benchmark targets a **relevant, interesting, and challenging capability**—**subtle comparative visual reasoning**—and highlighted three main strengths: (1) a **meaningful task definition** for subtle comparative reasoning that current VLMs still struggle with (Waq7, chNu, Qbbf), (2) **broad and diagnostic coverage** across diverse difference types and domains (Waq7, WLtr), and (3) **systematic experimental analysis** spanning fusion methods, prompting strategies, and controlled difficulty settings (chNu, Waq7, Qbbf).

To address reviewers’ remaining concerns regarding novelty, practical relevance, and improvability, we conducted several new experiments:

1. **Novelty and real-world relevance.** New correlation and transfer studies on industrial and aerial domains show that our benchmark aligns with downstream tasks more strongly than prior benchmarks. These analyses showed that performance on our benchmark exhibits a **higher correlation** with downstream accuracy, and fine-tuning on a small dataset built with our pipeline yields **larger downstream gains** than fine-tuning on existing benchmarks. These results demonstrate that the subtle comparative skills measured by our benchmark are more crucial to real-world VLM capability.
2. **Improving subtle comparative reasoning via fine-tuning.** We curated an additional training split and fine-tuned an open-source VLM. The model showed consistent improvements across all difference types, suggesting that our construction pipeline provides a clear path for strengthening this capability.
3. **Broader experimental coverage.** We added evaluations of **multi-image-native models** and newly proposed **prompting strategies**, including highlighting change regions and multi-step reasoning procedures. These experiments show that simple prompting is insufficient to mitigate the difficulty of subtle comparative reasoning.

We believe these additions meaningfully strengthen the contribution and directly address the reviewers’ main questions.

---

### Meta-Review · Area_Chair_pwL9 · 2026-01-02

**Summary:**

This paper introduces VLM-SubtleBench, a comprehensive benchmark designed to evaluate Vision-Language Models (VLMs) on the challenging task of subtle comparative reasoning across diverse visual domains and difference types. The reviewers recognized the submission as addressing a significant and underexplored problem, with notable strengths in its systematic dataset construction, broad coverage of domains and difference categories, and thorough diagnostic evaluation of both open-source and proprietary models. The benchmark reveals a substantial and interpretable performance gap between current VLMs and human-level capability, particularly in categories requiring spatio-temporal and geometric reasoning.

During the review and rebuttal process, the authors effectively addressed the reviewers' primary concerns. They provided substantial new evidence to demonstrate the real-world relevance and novelty of the benchmark through correlation and fine-tuning studies with downstream tasks in industrial anomaly detection and aerial surveillance. These experiments showed that performance on VLM-SubtleBench has a stronger correlation with real-world task performance and leads to greater fine-tuning gains compared to prior benchmarks, underscoring its practical value beyond a mere extension of existing work.

The authors also strengthened the paper by expanding the experimental analysis as suggested by the reviewers. This included evaluations of multi-image-native models, investigations of advanced prompting strategies (e.g., region highlighting, multi-step reasoning), a fine-tuning study demonstrating consistent improvements, and deeper controlled analyses of failure modes across difficulty levels and perceptual factors like color sensitivity.

While initial reviews raised questions about novelty, potential data artifacts, and the depth of analysis, the authors' comprehensive and evidence-based response convincingly clarified these points. Given the paper's contribution to the community in the form of a new benchmark and its insightful analysis, it is recommended for acceptance.

**Reviewer Concerns:**

Concerns Effectively Addressed by the Rebuttal:
1. Reviewer Waq7: Novelty and real-world relevance (W1).

Concern: The benchmark was perceived as a small increment over prior work, with subtlety defined via a potentially insufficient metric (DINOv3 similarity).

Response Addressed: The authors clarified that subtlety is a semantic and perceptual concept, with DINOv3 used only as a filtering tool. Crucially, they provided new correlation studies (with MMAD and QAG-360K) and fine-tuning transfer experiments showing that VLM-SubtleBench has a stronger correlation with downstream task performance and yields larger real-world gains than prior benchmarks (MLLM-CompBench). This strongly substantiated the benchmark's novel practical value.

2. Reviewer Waq7: Technical clarity on "overlap" and "subtraction" images (W4).

Concern: The method for creating these helper images was vague.

Response Addressed: The authors explicitly clarified that both operations are performed in pixel space, not feature space, and described the process (weighted blending for overlap, absolute difference for subtraction).

3. Reviewer Waq7 & chNu: Lack of evaluation on multi-image-native models (W5, related to W1).

Concern: Models like LLaVA-Next were not evaluated.

Response Addressed: The authors added evaluations of LLaVA-Next and LLaVA-OneVision, providing comparative results that show these models also struggle and generally underperform compared to other evaluated VLMs on this task.

4. Reviewer chNu: Suggestions for additional prompting strategies (W2).

Concern: The paper could explore more advanced baselines like region highlighting or two-step reasoning.

Response Addressed: The authors implemented and reported results for both suggested strategies. They found these methods offered only marginal or even negative impacts, reinforcing the paper's core finding that simple prompting is insufficient for this challenging task.

5. Reviewer Qbbf: Need for evidence of transfer to high-stakes domains (Q2).

Concern: The motivation mentions real-world applications but lacks proof of relevance.

Response Addressed: The authors provided the extensive correlation and fine-tuning studies (with MMAD and QAG-360K) as detailed above, directly demonstrating transfer and practical utility.

6. Reviewer Qbbf: Suggestion to extend synthetic analysis to color sensitivity (Q1).

Concern: The controlled study could probe known VLM color deficits.

Response Addressed: The authors conducted a new color-sensitivity analysis, confirming known weaknesses (e.g., with green and magenta) and showing these deficits compound comparative reasoning failures, adding a valuable diagnostic dimension.

Concerns Partially Addressed or Where Underlying Issues May Remain:
1. Reviewer Waq7: Potential lack of control in video-sourced pairs (W2/Q1).

Concern: For video frames, other elements may move besides the target, complicating the "subtle" comparison.

Response Addressed: The authors reframed this not as a flaw but as a deliberate design choice for ecological validity and increased difficulty, arguing it better reflects real-world conditions (e.g., surveillance). This is a reasonable philosophical justification.

Potential Outstanding Issue: A reviewer or reader might still feel this introduces a confounder, making it harder to isolate whether a model fails at the core comparative task or is simply distracted by irrelevant changes. The benchmark may be evaluating "subtle reasoning in noisy contexts" rather than "pure subtle reasoning."

2. Reviewer Waq7: Repetition of datasets across categories (W3).

Concern: The same source dataset (MYTEC-AD) is used for both "Attribute" and "State" categories.

Response Addressed: The authors clarified that while the source is the same, the specific image pairs, defects, and questions do not overlap, as they tap into different anomaly types (e.g., discoloration vs. physical damage).

Potential Outstanding Issue: Although logically separated, the reuse of a single data source for two fundamental difference types could be seen as a limitation in the breadth of data provenance for those categories, potentially affecting the generality of findings for "Attribute" and "State."

Overall Assessment:
The authors' rebuttal was highly effective and comprehensive. They directly and convincingly addressed the most critical concerns regarding novelty, relevance, and experimental breadth with substantial new evidence. The remaining points are largely matters of design philosophy (e.g., noisy video frames) or minor limitations in data sourcing, which do not undermine the paper's core contributions. Therefore, the decision is to accept the paper.

**Reviewer Scores:**

Reviewer Waq7 (Initial: 2 - Reject): The authors convincingly countered the major criticism of lack of novelty by providing strong evidence of higher correlation with downstream tasks and greater transfer gains than prior benchmarks. This directly addressed the core weakness. The reviewer might have acknowledged this significant improvement while possibly retaining some reservations about video control. Their score would likely increase from 2 to 4.

Reviewer chNu (Initial: 6 - Marginally Above Threshold): The authors implemented the requested experiments on prompting and fine-tuning, which validated and strengthened the paper's conclusions. The reviewer was already positive, and these additions would have solidified their support. Their score would likely remain a 6.

Reviewer Qbbf (Initial: 6 - Marginally Above Threshold): The authors provided the exact transfer evidence and deeper diagnostic analysis (color sensitivity) requested. These were high-quality additions that met the reviewer's conditions for acceptance. Their score would likely remain a 6.

Reviewer WLtr (Initial: 8 - Accept): Already the strongest supporter, the author's addition of fine-tuning analysis and discussion on failure modes would have reinforced their positive view. Their score would remain an 8.

Conclusion: The rebuttal effectively addressed major concerns, likely moving the most critical reviewer (Waq7) from rejection to the borderline, while solidifying the positive or cautiously positive scores of the others. This convergence supports the acceptance decision.

---

### Decision · Program_Chairs · 2026-01-26

Accept (Poster)